# Spatial Distribution of Oceanic Moisture Contributions to Precipitation over the Tibetan Plateau

Ying Li[1,2], Chenghao Wang[3,4,5], Ru Huang[6], Denghua Yan[7], Hui Peng[1,2], and Shangbin Xiao[1,2]

[1]Engineering Research Center of Eco-environment in Three Gorges Reservoir Region, Yichang 443002, China;
[2]College of Hydraulic and Environmental Engineering, China Three Gorges University, Yichang, 443002, China
[3]Department of Earth System Science, Stanford University, Stanford, CA 94305, USA
[4]School of Meteorology, University of Oklahoma, Norman, OK 73072, USA
[5]Department of Geography and Environmental Sustainability, University of Oklahoma, Norman, OK 73019, USA
[6]Ministry of Emergency Management of China, National Institute of Natural Hazards, Beijing 100085, China
[7]State Key Laboratory of Simulation and Regulation of Water Cycle in River Basin, Water Resources Department, China Institute of Water Resources and Hydropower Research (IWHR), Beijing, 100038, China

*Correspondence to*: Ying Li (ly_hyrdo@outlook.com); Hui Peng (hpeng1976@163.com)

**Abstract.** Evaporation from global oceans is an important moisture source for glaciers and headwaters of major Asian rivers in the Tibetan Plateau (TP). Although the accelerated global hydrological cycle, the altered sea–land thermal contrast, and the amplified warming rate over the TP during the past several decades are known to have profound effects on the regional water balance, the spatial distribution of oceanic moisture contributions to the vast TP remains unclear. This hinders the accurate quantification of regional water budgets and the reasonable interpretation of water isotope records from observations and paleo archives. Based on historical data and moisture tracking, this study systematically quantifies the absolute and relative contributions of oceanic moisture to long-term precipitation in the TP. Results show that the seasonal absolute and relative oceanic contributions are generally out of phase, revealing the previously underestimated oceanic moisture contributions brought by the westerlies in winter and the overestimated moisture contributions from the Indian Ocean in summer. Quantitatively, the relative contribution of moisture from the Indian Ocean is only ~30% in the south TP and further decreases to below 10% in the northernmost TP. The absolute oceanic contribution exhibits a spatial pattern consistent with the dipole pattern of long-term precipitation trends across the Brahmaputra Canyon region and the central-northern TP. In comparison, relative oceanic contributions show strong seasonal patterns associated with the seasonality of precipitation isotopes across the TP.

## 1 Introduction

Evaporation from oceans is one of the most important elements in the global hydrological cycle, which constitutes more than 80% of the global surface evaporation and contributes to about half of the terrestrial precipitation (*Van der Ent et al.*, 2010; *Trenberth et al.*, 2011; *Gimeno et al.*, 2020a; *Link et al.*, 2020; *Tuinenburg et al.*, 2020). The global-warming-induced increase of water-holding capacity in the atmosphere, intensification of the land-sea thermal gradient, and the relevant

moisture limitation over land (i.e., soil moisture limitation) collectively enhance the role that oceanic evaporation plays in the global hydrological cycle (*Findell et al.*, 2019; *Algarra et al.*, 2020; *Gimeno et al.*, 2020b). Owing to the complex circulation systems that involve energy-intensive processes, large spatial and temporal variations in oceanic evaporation (and its contributions to precipitation over land) have been observed at the regional scale (*Gimeno et al.*, 2013; *Van der Ent and Savenije*, 2013). Specifically, the continental regions influenced by monsoon systems have been considered to substantially benefit from oceanic evaporation (*Gimeno et al.*, 2010). One example is the high-elevation Tibetan Plateau (TP) (*Yao et al.*, 2012; *Yao et al.*, 2013; *Yao et al.*, 2018). Although located far from oceans, the TP has long been considered a gigantic "air pump" that attracts low-latitude oceanic moisture up to the Asian continent, resulting from its large-scale topography and thermal forcing (*Xu et al.*, 2014; *Wu et al.*, 2015; *Liu et al.*, 2020). More importantly, the TP region sustains freshwater supplies for more than 10 major Asian rivers affecting billions of livelihoods downstream (*Immerzeel et al.*, 2010; *Lutz et al.*, 2014). In the context of global change, the TP region is undergoing dramatic hydrological changes (e.g., the intensive cryosphere melt and lake expansion) (*Yao et al.*, 2012; *Zhang et al.*, 2020). Meteorological records reveal that the atmospheric warming rate over the TP was twice the global mean during the past five decades, and experienced further accelerations since 1998 (*Chen et al.*, 2015; *Duan and Xiao*, 2015; *Kuang and Jiao*, 2016). The consequent changes in the huge land–sea thermal gradient may have significantly altered regional moisture transport processes and circulation systems (*Wang et al.*, 2019).

Hydrological conditions in different parts of the TP are closely connected through the interaction between the mid-latitude westerlies and the Indian Summer Monsoon (ISM) as well as the strong local recycling (*Xu et al.*, 2008; *Yao et al.*, 2013; *Curio and Scherer*, 2016). During the monsoon season (June–September), regional heating significantly enhances the south-westerly monsoon circulations over the north Indian Ocean, which brings enormous oceanic moisture to South Asia and the TP (*Xu et al.*, 2008; *Wu et al.*, 2015). However, the impacts of the ISM have gone through changes during recent decades. The rapid Indian Ocean warming during the 20[th] century has potentially weakened the land–sea thermal contrast, which dampened the summer monsoon Hadley circulation (*Bingyi*, 2005; *Roxy et al.*, 2015). After 2002, the increased land–ocean temperature gradient driven by the enhanced warming of the Indian subcontinent and the slowed warming of the Indian Ocean resulted in the revival of the ISM (*Jin and Wang*, 2017). These changes may have altered the oceanic moisture contributions to precipitation over the TP. In fact, the hinterland TP (mainly the central-northern TP) has become wetter in the recent 50 years, while a drying trend has been observed near the southeastern edge of the TP (*Yang et al.*, 2014; *Jiang and Ting*, 2017; *Wang et al.*, 2018).

Many recent studies have quantitatively diagnosed the oceanic moisture contributions for different climate regions of the TP (Table S1). As suggested in Table S1, the vast majority the studies have investigated the moisture sources of precipitation in the TP at both regional and subregional scales ("Study area" in Table S1) using backward moisture tracking. However, the

spatial distribution of oceanic moisture contributions to precipitation over vast TP, e.g., the potential latitudinal gradient of the moisture transported from the Indian Ocean, has not been examined.

On the other hand, the accurate quantification of oceanic evaporation contributions can also benefit the interpretation of water isotopes and paleoclimate archive records (i.e., ice core, tree ring, lake sediments, and stalagmites) gathered in the TP
over the past several decades (*Tian et al.*, 2007; *Joswiak et al.*, 2013; *Yao et al.*, 2013; *Zhu et al.*, 2015; *Kumar et al.*, 2021). Water isotopes—the stable isotopic compositions of hydrogen and oxygen—have been widely used to examine the climate and water cycle over the TP, including their paleoclimate history (*Joswiak et al.*, 2013; *Yao et al.*, 2013). Specifically, extensive evidence from precipitation and ice core isotopes since the 1990s has demonstrated that the onset of the ISM delivers substantial oceanic moisture to the TP as far as the Tanggula Mountains (34°–35°N) (*Tian et al.*, 2007; *Yao et al.*,
2013). Nevertheless, a quantitative understanding of the relationship between the moisture sources and the spatial-temporal variations in water isotopes is still absent.

In addition, the moisture contributions to precipitation over a target region can be viewed from two aspects: absolute and relative contributions. Although relative contribution can be calculated from the absolute contribution, these two metrics in
fact reflect different aspects of the regional hydrological cycle. The absolute contribution, which is critical to the water balance, is critical to the understanding of the hydrological status and its dynamics. In comparison, from the perspective of mass balance during mixing, the relative contribution is more relevant to tracer-based studies. However, the differences between absolute and relative contributions have rarely been explored over the TP (*Zhang et al.*, 2017; *Pan et al.*, 2018; *Chen et al.*, 2019; *Qiu et al.*, 2019).


In this study, we aim to fill these gaps with long-term moisture tracking simulations driven by multiple reanalysis datasets for the TP. In sections 3.1 and 3.2, we quantify the spatial variations in the absolute and relative contributions of the oceanic moisture to the TP precipitation. We then compare the long-term trends of oceanic moisture changes and precipitation changes in Section 3.3. We further examine the possible influence of oceanic moisture on the variations in water isotope
records over the TP. Leveraging systematic forward and backward moisture tracking simulations, the results of this study are expected to shed new light on the oceanic impacts and dynamics of the hydrological cycle in the TP.

## 2 Method and data

### 2.1 Numerical atmospheric moisture tracking

The Water Accounting Model-2layers (WAM-2layers) is an Eulerian posterior moisture tracking model which can track
tagged moisture both forward and backward in time to determine the spatial and temporal distributions of moisture sources (*Van der Ent et al.*, 2010; *Van der Ent*, 2014). In comparison with the commonly used Lagrangian models (e.g., the

FLEXible PARTicle (FLEXPART) dispersion model and the Hybrid Single-Particle Lagrangian Integrated Trajectory (HYSPLIT) model) that concern the movement of "air particles" in the atmosphere and identify precipitation and evaporation events mainly based on the dynamic humidity information of the tracked particles, Eulerian models principally

focus on moisture transport among fixed grids. In general, Lagrangian models are more accurate and run faster than Eulerian ones for short-term moisture tracking of single grid cells, while Eulerian models are more efficient for long-term moisture tracking over large target regions (*Tuinenburg and Staal*, 2020). More importantly, the selection of WAM-2layers enables us to consider moisture budget from precipitation and evaporation separately on Eulerian grids (*Van der Ent et al.*, 2013; *Van der Ent*, 2014).


The basic principle of the forward moisture tracking for the tagged moisture (subscript $g$) in WAM-2layers in the lower layer is the atmospheric water balance (*Findell et al.*, 2019):

$$\frac{\partial S_{g,lower}}{\partial t} = -\frac{\partial (S_{g,lower} u)}{\partial x} - \frac{\partial (S_{g,lower} v)}{\partial y} + E_g - P_g \pm F_{v,g} \qquad (1)$$

where $S_g$ is the moisture storage in the atmospheric column; t is time; $u$ and $v$ are wind speeds in the zonal (x) and

meridional (y) directions, respectively; $E_g$ and $P_g$ are evaporation entering and precipitation leaving the layer, respectively; and $F_v$ is the vertical moisture exchange between the lower and upper layers. Note that $E_g$ only applies to the lower layer. The "well-mixed" assumption is adopted in this model, which means that precipitation is assumed to be immediately removed from the atmosphere in the tracking process (i.e., $P_g/P = S_g/S$, where $P$ and $S$ are total precipitation and total column atmospheric moisture storage, respectively). The two vertical layers in the model are set to deal with the wind shear

in the upper air. To better capture the vertical exchanges due to convection, turbulence, and re-evaporation and to minimize the water balance losses between the two layers, the gross vertical flow is set to 4 times the vertical flow in the net flow direction and 3 times the vertical flow in the opposite direction. Although this is a simplification of the turbulent moisture exchange, physically reasonable results have been obtained in previous studies, and the general tracking has been validated against the online 3D tracking models (*Van der Ent et al.*, 2013; *van der Ent et al.*, 2014; *Findell et al.*, 2019).


In the tracking process, the spatial resolution of the Eulerian grids is reduced to 1°×1°, and the time step is set as 0.25 h to maintain precision and numerical stability. We have tested the sensitivity of the moisture tracking results to the selection of different time steps. Figure S1 shows the comparison of two simulations using 15-min (0.25 h) and 10-min time steps, suggesting the stability of using different time steps for moisture tracking in the study area. In addition, the vertical

separation between the two layers is prescribed as ~812 hPa at the normal atmospheric pressure (*Van der Ent et al.*, 2013). Note that the atmospheric pressure of the vertical separation varies with different surface pressure (the "half-level" pressure in different reanalysis products is defined as $P_{k-1/2} = A_{k-1/2} + B_{k-1/2} P_s$, where $P_s$ is surface pressure, k represents different model levels, and the values of $A_{k-1/2}$ and $B_{k-1/2}$ are defined independently for different reanalysis datasets). Based on our

preliminary experiments, we select a tracking domain that covers nearly all the potential ocean and land source regions of the

TP precipitation (30°S–80°N and 40°W–140°E).

In an Eulerian grid cell $(x, y)$ at time $t$, the relative contribution (%) of oceanic evaporation to precipitation is defined as:

$$\rho_o(t, x, y) = \frac{M_o(t,x,y)}{M_o(t,x,y)+M_l(t,x,y)} \tag{2}$$

where $M_o$ is the absolute contribution of oceanic moisture, and $M_l$ is the absolute contribution of terrestrial moisture

(including local recycling). The ocean and land distributions were defined according to the 1°×1° gridded land–sea mask from the ERA-Interim. We further remove inland large lakes (considered as "sea" in the ERA-Interim, e.g., the Caspian Sea and the Black Sea in the Eurasian continent) from the mask. The final land–sea mask with 1°×1° spatial resolution used in this study is shown in Figure S2.

## 2.2 Data source

Three atmospheric reanalysis products, namely the European Centre for Medium-Range Weather Forecasts (ECMWF) interim reanalysis dataset (ERA-Interim) (*Dee et al.*, 2011), the National Aeronautics and Space Administration Modern-Era Retrospective Analysis for Research and Applications version 2 dataset (MERRA-2) (*Gelaro et al.*, 2017), and the Japanese 55-year Reanalysis dataset (JRA-55) (*Kobayashi et al.*, 2015), are used to drive the WAM-2layers. Variables used in our simulations include surface pressure (temporal resolution: 6h), precipitation (3h), evaporation (3h), specific humidity (6h),

wind fields (6h), total column water (6h), and vertically integrated moisture fluxes (6h). Note that for specific humidity and wind fields, 17 model layers are selected to represent the moisture distribution from the surface to the top of the atmosphere (see details in Table S2). To better describe the water transport in the atmosphere, the moisture considered in this study represents all possible phases of water in the atmosphere, including water vapor, cloud liquid water, and cloud frozen water. Note that JRA-55 does not provide liquid and frozen water fluxes, thus we only consider water vapor flux for this dataset.

The time span of moisture tracking is 1979–2015 for ERA-Interim and JRA-55, and 1980–2015 for MERRA-2. All variables are temporally resampled to 0.25 h and spatially interpolated to 1°×1° grids using bilinear interpolation for consistency.

We further retrieve the 1°×1° gridded monthly averaged data from the ERA5 (the fifth generation of atmospheric reanalysis product produced by the ECMWF) (*Hersbach et al.*, 2020) to examine the large-scale changes in evaporation, precipitation,

horizontal wind fields, and vertical velocity in the upper atmosphere during 1979–2015. The event-based precipitation $\delta^{18}O$ data from 19 observation stations of the Tibetan Network for Isotopes in Precipitation (TNIP) are also used in this work (*Yao et al.*, 2013).

## 3 Results

### 3.1 Absolute and relative contributions of oceanic moisture to the TP precipitation

The absolute contribution (mm per year, season, or month) and relative contribution (the percentage of total sink precipitation) of global oceanic evaporation to the TP precipitation over the past three decades are tracked forward using an Eulerian moisture tracking model (WAM-2layers) and three reanalysis products (ERA-Interim, MERRA-2, and JRA-55). The spatial patterns of absolute contributions (Figure 1a–e) agree well with the previous understanding that the ISM brings a large amount of Indian ocean moisture to the southeastern TP and concentrates around the Brahmaputra Canyon region in

summer (*Xu et al.*, 2014; *Wu et al.*, 2015). In the monsoon season (summer, Figure 1c), the absolute contribution of oceanic moisture exhibits a sudden drop from more than 1,000 mm along the southern TP to only about 100 mm in the central-northern TP, after traveling through the orographic barriers of the Gangdise and the southern slope of the Tanggula Mountains. This massive oceanic moisture also stretches westward along the southwestern slope of the Himalayas. In comparison, in spring and winter (Figure 1b and e), a relatively weak westerly moisture sink appears around the Pamirs

(~100 mm season$^{-1}$), as induced by the prevailing orographic precipitation (*Curio and Scherer*, 2016).

    From the perspective of relative contribution, oceanic evaporation is responsible for 36%–39% of the total moisture condensation (precipitation) over the TP, and spatially, the relative contribution gradually decreases from more than 50% along the southeastern edge of the TP to less than 20% in the central-northern TP (Figure 1f). Seasonally, the relative

contributions of oceanic evaporation to the TP precipitation are 33%–41%, 36%–39%, 35%–38%, and 51%–54% in spring, summer, autumn, and winter, respectively. In spring and autumn (Figure 1g and i), the relative contribution decreases gradually from the southern and southwestern TP to the central-northern TP. In summer (Figure 1h), the relative contributions exhibit roughly a latitudinal distribution, with the 50% isoline located between 25°N and 30°N and the 30% isoline between 30°N and 35°N. This pattern is distinct from that of the absolute contribution featured by oceanic moisture

concentrating around the southeastern TP. It is also notable that the westernmost TP (the Pamir region) shows the lowest relative contribution from oceans in summer in comparison with that in other seasons. This indicates a more active role that terrestrial evapotranspiration plays in the western TP during the monsoon season.

    The largest contrast between the absolute and relative contributions occurs in winter when the absolute contribution reaches

the lowest level while the relative contribution peaks (Figure 1e and j). Based on the backward tracking of seasonal precipitation sources for the entire TP (Figure S5), in winter, the moisture contribution from the westerlies-dominated oceans (the Mediterranean, the Red Sea, the Persian Gulf, and even the Atlantic) is much higher than that from the cold and dry Eurasian continent (Figure S5d–f). Although the Mediterranean, the Red Sea, and the Persian Gulf are much smaller than the Atlantic, their moisture contribution to the TP precipitation is non-negligible. In fact, the total contributions of these three

oceanic source regions can be greater than that of the Atlantic during both the monsoon and non-monsoon seasons (Table

S3). In winter, the westerlies push the 50% isoline of the relative oceanic contribution eastward to the mid-eastern TP, with the relative contribution well above 30% for the entire TP. The spatial patterns on annual and seasonal scales observed here are also consistent among simulations driven by different reanalysis datasets (Figures S3 and S4). Figure 2a shows a thorough comparison of monthly oceanic moisture contributions to precipitation over the entire TP among different simulations. The absolute and relative oceanic contributions are in general out of phase: high absolute (relative) contributions in summer (winter) and low absolute (relative) contributions in winter (summer), no matter which reanalysis dataset is used.

## 3.2 Moisture contributions from different oceans

The backward tracking of precipitation for the entire TP shows that the moisture sources could extend far west to the Atlantic as driven by the mid-latitude westerlies and far south to the southern Indian Ocean as dominated by the monsoon system (Figure S6a–c). Given the importance of the interactions between the westerlies and the ISM in determining the TP's climate and hydrologic cycle (*Xu et al.*, 2008; *Yao et al.*, 2013; *Curio and Scherer*, 2016; *Yao et al.*, 2018), we divide the major oceanic source regions into the western oceans part (WO, including the Mediterranean, the Red Sea, the Persian Gulf, and the eastern Atlantic) and the Indian Ocean part (IO), as shown in Figure S6d.

Annually, the absolute contribution of moisture from the WO decreases from above 100 mm along the western and southern TP to around 20 mm in the central and northeastern TP (Figure 3a). Seasonally (Figure 3b–e), in addition to most parts of the western TP during the non-monsoon seasons, the southwestern edge of the TP, particularly in spring and summer, is also substantially influenced by the moisture transported from the WO. In comparison, the relative contribution gradually weakens from the northwest to the southeast of the TP on both annual and seasonal scales (Figure 3f–j). This is consistent with the prevailing orographic precipitation dominated by westerly moisture transport and the zonal movement of the westerlies (*Curio and Scherer*, 2016). However, the relative contribution of the WO drops to below 10% over the entire TP in summer, because the outbreaking of the ISM and the enhanced evapotranspiration from the wetting Eurasian continent dominate the available moisture over the TP (Figure S5a–c). For the monthly variations in the WO moisture contribution to the TP precipitation (Figure 2b), the absolute and relative contributions exhibit a phase shift of about three months, with the high absolute (relative) contribution in spring (winter) and the low absolute (relative) contribution in autumn (summer).

Considering both absolute and relative contributions to the regional precipitation, the moisture contribution of the IO is significantly higher than that of the WO in most parts of the TP, except for the northwesternmost TP. Along with the onset and retreat of the ISM, the absolute contribution from the IO exceeds 500 mm during summer in the southeasternmost TP and falls to below 100 mm during winter nearly over the entire TP (Figure 4b–e). The relative contribution from the IO exhibits roughly a latitudinal gradient on annual scale (Figure 4f), with the 30% and 10% isolines located in the southern and northern TP, respectively. This zonal pattern lasts from spring to autumn (Figure 4g–i), indicating a dynamic balance

between the IO moisture contribution and the total moisture convergence during the period. This synchronism is largely broken in winter when the highest level of the relative contribution from the IO shifts to the southwesternmost TP (Figure 4j), with most IO moisture from the Arabian Sea (Figure S5d–f). In fact, in the westerlies-dominated winter, the IO is still the major oceanic moisture source of precipitation in the southern TP, while the WO is the dominating oceanic moisture source for the western and northern TP. For the monthly variations of the IO moisture contribution (Figure 2c), the absolute contribution reaches its maximum in summer owing to the significantly enhanced ISM, whereas the relative contribution peaks in both summer and winter. Additional simulations based on MERRA-2 and JRA-55 datasets both suggest similar spatial patterns of moisture contribution from different oceans (Figures S7–10).

### 3.3 Long-term trends of oceanic moisture contributions to the TP precipitation

Figure 5 shows the annual trends of oceanic moisture contributions to the TP precipitation during 1979/1980–2015 estimated based on three simulations (driven by ERA-Interim, MERRA-2, and JRA-55). Despite some spatial discrepancies, all three simulations reveal a rough dipole pattern spanning from the southeastern TP to the central-northern TP. The most notable area is the Brahmaputra Canyon region, the most important moisture transport channel for the TP (*Hren et al., 2009*), which has gone through the most significant decrease in oceanic moisture contributions. Specifically, the moisture contributions from both the monsoon-dominated IO and the westerlies-dominated WO show decreasing trends in the Brahmaputra Canyon region (Figure S11). Despite the enhanced evaporation from most oceanic source regions during 1979–2015 (Figure 6a), this oceanic moisture may loss significantly due to the increased precipitation sink along the moisture transport pathway before reaching the target region, for example, when traveling across the Indian Subcontinent and the Bay of Bengal (Figure 6b). In addition, we observe significantly weakened eastward and northward winds in the lower atmosphere (700 hPa) around the Brahmaputra Canyon (Figure 7), suggesting that less moisture may transport through the region in the lower atmosphere. Meanwhile, we also observe significantly weakened upward motion in the region, which further verifies the weakened moisture convergence around the Brahmaputra Canyon region.

Regional precipitation is fuelled by moisture that is either transported directly from oceans or recycled from lands (*Gimeno et al.*, 2020a). In most cases, oceanic moisture contributes to less than 50% of the total precipitation in the TP (except for the southeasternmost TP in summer and the western TP in winter; Figure 1). Compared with the relative contribution of oceanic moisture, the absolute contribution of oceanic moisture (especially that from the IO) exhibits a spatial pattern highly consistent with precipitation distributions over the TP on both annual and seasonal scales (Figure S12). Previous studies have demonstrated that the precipitation seasonality over the TP is determined jointly by the zonal movement of the westerlies and the onset and retreat of the ISM, suggesting the potential connections between oceanic moisture contributions and precipitation dynamics (*Xu et al.*, 2014; *Curio et al.*, 2015; *Yao et al.*, 2018). Indeed, the long-term (1979–2015) trends of precipitation and oceanic moisture contributions have similar dipole patterns (e.g., decreased precipitation in the Brahmaputra Canyon region but increased precipitation in the central-northern TP) (Figures 5 and 6b). To examine whether

the decreased precipitation around the Brahmaputra Canyon region is mainly due to the changes in oceanic moisture contributions, we carried out additional backward moisture tracking simulations for the southeastern TP region (SETP) (Figure S13a). The SETP defined here roughly covers areas dominated by the decrease in both precipitation and absolute

oceanic moisture contribution (the purple rectangle in Figure 5). As shown in Figure S13b, the moisture contributions of both the westerlies-dominated western sources and the ISM-dominated southern sources to the SETP decreased over time, and most source regions with substantial decreases are over land. Meanwhile, only few areas in the southwestern slope of the Himalayas and the southwestern corner of the TP show enhanced moisture contribution to precipitation in the SETP. Actually, *Jiang and Ting* (2017) have suggested that the dipole precipitation pattern over the TP are likely attributable to the

interactions between the ISM and the TP. Nevertheless, more thorough analyses of the land–atmosphere interactions with physics-based models at higher spatial resolution are still needed to better understand these interactions and their response to climate change.

## 3.4 Relative contributions of oceanic moisture associated with the patterns of water isotopes

Based on numerous precipitation isotopic observations and ice-core records since the 1990s, previous studies have identified

three distinct climate regions in the TP, as governed by the westerlies (northern TP; hereafter westerlies domain), the ISM (southern TP; hereafter monsoon domain), and their interactions (hereafter transition domain) (*Tian et al.*, 2007; *Joswiak et al.*, 2013; *Yao et al.*, 2013). Theoretically, moisture delivered by the ISM tends to have relatively low isotope values due to strong convection activities along its transport paths, whereas moisture delivered by the westerlies in general has relatively high isotope values (*Bowen et al.*, 2019; *Cai and Tian*, 2020). Here we further investigate the relationships between the

simulated oceanic moisture contributions and precipitation $\delta^{18}O$ observations at 19 monitoring stations over the TP (Figures 8 and S14–17).

Compared with other oceanic moisture contributions, the relative contribution of oceanic moisture from the IO is strongly correlated with precipitation $\delta^{18}O$ (Figure 8), and nearly all stations show negative correlations (Table S4). In the monsoon

domain, $\delta^{18}O$ is high in spring and low in summer (note that the $\delta^{18}O$ axes are revised), and correspondingly, the relative contribution from the IO is low in spring and high in summer. In comparison, in the westerlies domain, high $\delta^{18}O$ is associated with low relative contributions from the IO during the monsoon season, while low $\delta^{18}O$ is associated with high relative contributions from the IO during the non-monsoon seasons. In the transition domain, the seasonal cycles of $\delta^{18}O$ and the relative contribution from the IO are similar to those in both the monsoon domain and the westerlies domain. Note the

mismatch between the summer peaks of the relative moisture contribution from the IO and the low $\delta^{18}O$ values in autumn at Lulang, Nuxia, and Bomi stations near the Brahmaputra Canyon, which is likely due to the impact of the moisture transport from Southeast Asia or the Pacific Ocean driven by the trough embedded in the south branch of the westerlies (*Cai and Tian*, 2020). Additional backward moisture tracking of the monthly moisture sources for the SETP (which covers Lulang, Nuxia,

and Bomi stations) also suggests that the moisture sources gradually extend to Southeast Asia and the western Pacific Ocean during September and October (Figure S18).

Except for several sites in the southernmost TP, our results confirm the theory that a higher percentage of oceanic moisture contribution from the ISM-dominated IO is associated with a lower precipitation $\delta^{18}O$ value over the TP. As for the relative contribution from the westerlies-dominated WO (Figure S15), the absolute contribution from the WO (Figure S16), and the absolute contribution from the IO (Figure S17), their relationships with the water isotope ratios are much weaker. Based on the seasonality of water isotope ratios, previous studies have identified a northern boundary around 34°–35°N for the impact of the ISM on the TP, where the Tanggula Mountains serve as a main orographic barrier (*Tian et al.*, 2007; *Joswiak et al.*, 2013; *Yao et al.*, 2013). Quantitatively, this geographical barrier reflected in water isotope ratios in general aligns with the 10%–20% (~20%) isoline of the relative contribution from the IO (entire ocean sources) in summer (Figure 4h).

## 4 Conclusions and discussion

From the perspectives of absolute and relative contributions, this work quantifies the oceanic evaporation contributions to precipitation over the TP region. After crossing the surrounding mountain ranges of the TP, the absolute (relative) contribution of moisture from global oceans rapidly decreases from more than 1000 mm (50%) around the Brahmaputra Canyon region to only about 100 mm (10%) in the central-northern TP. However, substantial variations in the spatial patterns exist on the seasonal scale. For example, the highest absolute contribution of oceanic evaporation to the TP precipitation occurs in the southeast during summer, while the highest relative contribution occurs in the western TP during winter. Previous studies primarily focused on oceanic moisture contributions to the TP precipitation from the monsoon-dominated Indian Ocean (*Xu et al.*, 2008; *Yao et al.*, 2013). In contrast, our results highlight that the westerlies-dominated oceans, such as the Mediterranean, the Red Sea, the Persian Gulf, and even the Atlantic, are also important source regions for the TP precipitation, especially during the non-monsoon seasons (e.g., contribution exceeds 30% in the westernmost TP in winter).

In addition, we found that the absolute contribution of oceanic moisture, when compared with relative contribution, is more consistent with precipitation in terms of spatial patterns, while the relative contribution to some extent reflects the variations of precipitation isotopes. The spatial pattern of trends in the absolute oceanic moisture contributions reflects the dipole pattern of precipitation change across the Brahmaputra Canyon region and the central-northern TP. Meanwhile, the seasonal variations of relative contribution from the IO are generally out of phase with the precipitation $\delta^{18}O$ over much of the TP in all three climate domains. We acknowledge that beyond the influence from moisture sources, precipitation is also collectively affected by multiple synoptic and climate factors and so are the precipitation isotopes (*Dansgaard*, 1964; *Galewsky et al.*, 2016; *Bowen et al.*, 2019). Nevertheless, this work systematically quantifies the oceanic moisture

contributions to the vast TP, and provides new insight into the influence of oceanic moisture contribution on water cycle and water isotope records in this region. Future studies on multi-source and multi-process moisture transport are expected to further enrich our understanding of the paleoclimate proxy records and global-warming-induced water resource changes over the TP—the "Asia water tower" and the core area of the Belt and Road Initiative.

## Data availability

The ERA-Interim dataset can be downloaded from the official website of the European Centre for Medium-Range Weather Forecasts (ECMWF): https://www.ecmwf.int/en/forecasts/datasets/reanalysis-datasets/era-interim (*ECMWF, 2017*). The MERRA-2 dataset is available at https://gmao.gsfc.nasa.gov/reanalysis/MERRA-2/ (*NASA Goddard Earth Sciences Data and Information Services Center, 2018*), which is managed by the Goddard Earth Sciences Data and Information Services Center (GES DISC), National Aeronautics and Space Administration (NASA). The JRA-55 product was developed by the Japan Meteorological Agency and can be downloaded from https://jra.kishou.go.jp/ (*Japan Meteorological Agency, 2018*). The ERA5 dataset can be downloaded from the Copernicus Climate Change Service (C3S) Climate Date Store (CDS): https://cds.climate.copernicus.eu/ (Copernicus Climate Change Service CDS, 2021). The TNIP $\delta^{18}$O data can be downloaded from the National Tibetan Plateau/Third Pole Environment Data Center: http://data.tpdc.ac.cn (*National Tibetan Plateau Data Center, 2021*). The code of WAM-2layers (v2.4.08) is available at https://github.com/ruudvdent/WAM2layersPython (*van der Ent, 2022*). The results of oceanic moisture tracking simulations are archived at the National Tibetan Plateau Data Center (TPDC): http://data.tpdc.ac.cn/en/data/c6f758cf-6c99-4023-8026-f59e6d3657cb/ (*Li et al., 2022*).

## Author contribution

YL and CW conceptualized the study. YL carried out numerical simulations, conducted formal analysis, prepared figures, and wrote the initial draft. YL, CW, and RH contributed to the visualization of results. All authors contributed to the review and editing of the manuscript.

## Competing interests

The authors declare that they have no conflict of interest.

## Acknowledgements

This work was financially supported by the Second Tibetan Plateau Scientific Expedition and Research Program (2019QZKK020705) and the Natural Science Foundation of Hubei Province of China (2022CFB785). We thank Dr. Zhongyin Cai for his contribution to the isotope analysis and valuable comments on the initial draft of this manuscript. We

thank Dr. Ruud van der Ent, Dr. Thom Bogaard, and an anonymous reviewer for their thorough review and constructive comments, which substantially improve our manuscript.

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

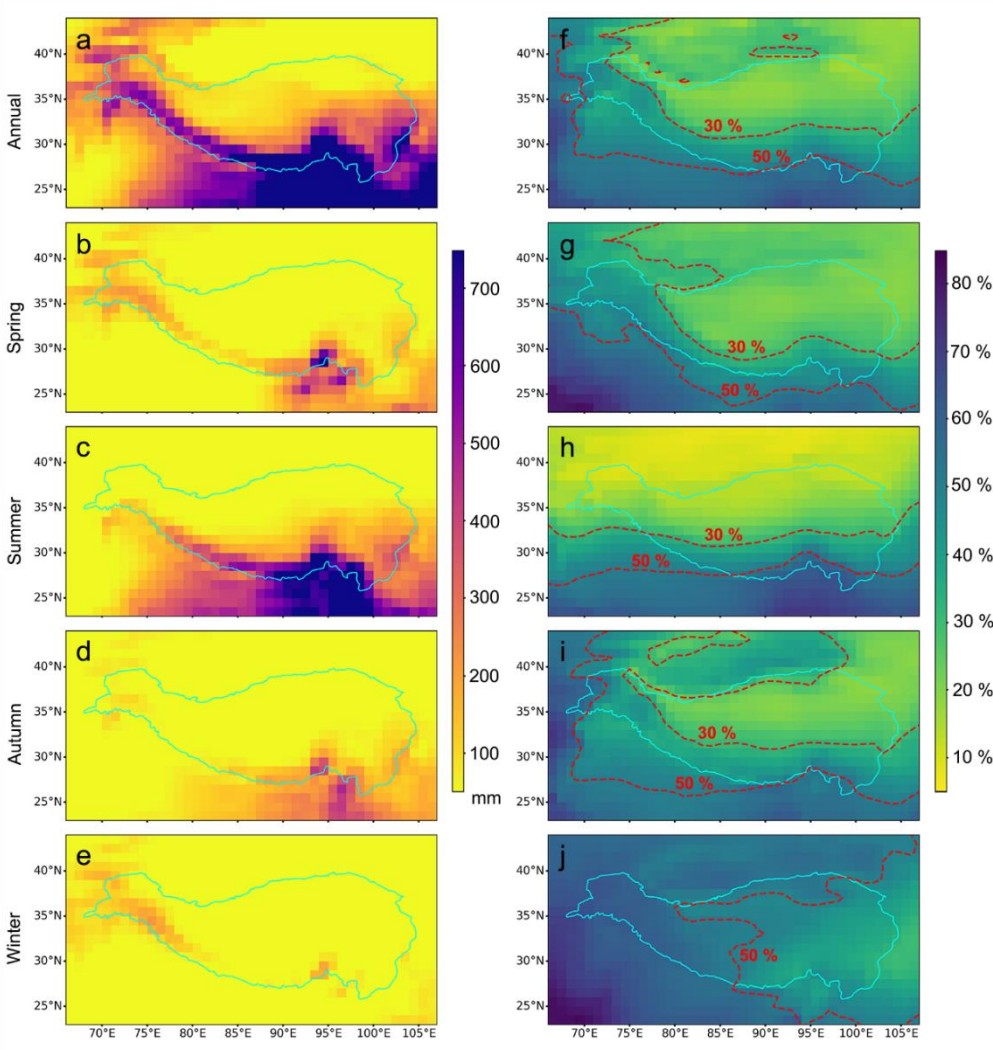

**Figure 1:** Spatial distributions of long-term mean absolute and relative oceanic moisture contribution to the TP precipitation. (**a**–**e**) The absolute contribution from global oceans (mm, equivalent water height) on annual (mm year$^{-1}$) and seasonal (mm season$^{-1}$) scales. (**f**–**j**) The relative contribution of oceanic moisture (%, the percentage of oceanic contribution relative to total moisture convergence) on annual and seasonal scales. Cyan lines represent the TP boundary, and dashed red lines in (**f**–**j**) are 30% and 50% isolines of the relative contribution. The forward moisture tracking results are modelled using WAM-2layers driven by ERA-Interim during 1979–2015. Moisture-tracking results driven by MERRA-2 (1980–2015) and JRA-55 (1979-2015) are shown in Figures S3 and S4, respectively.

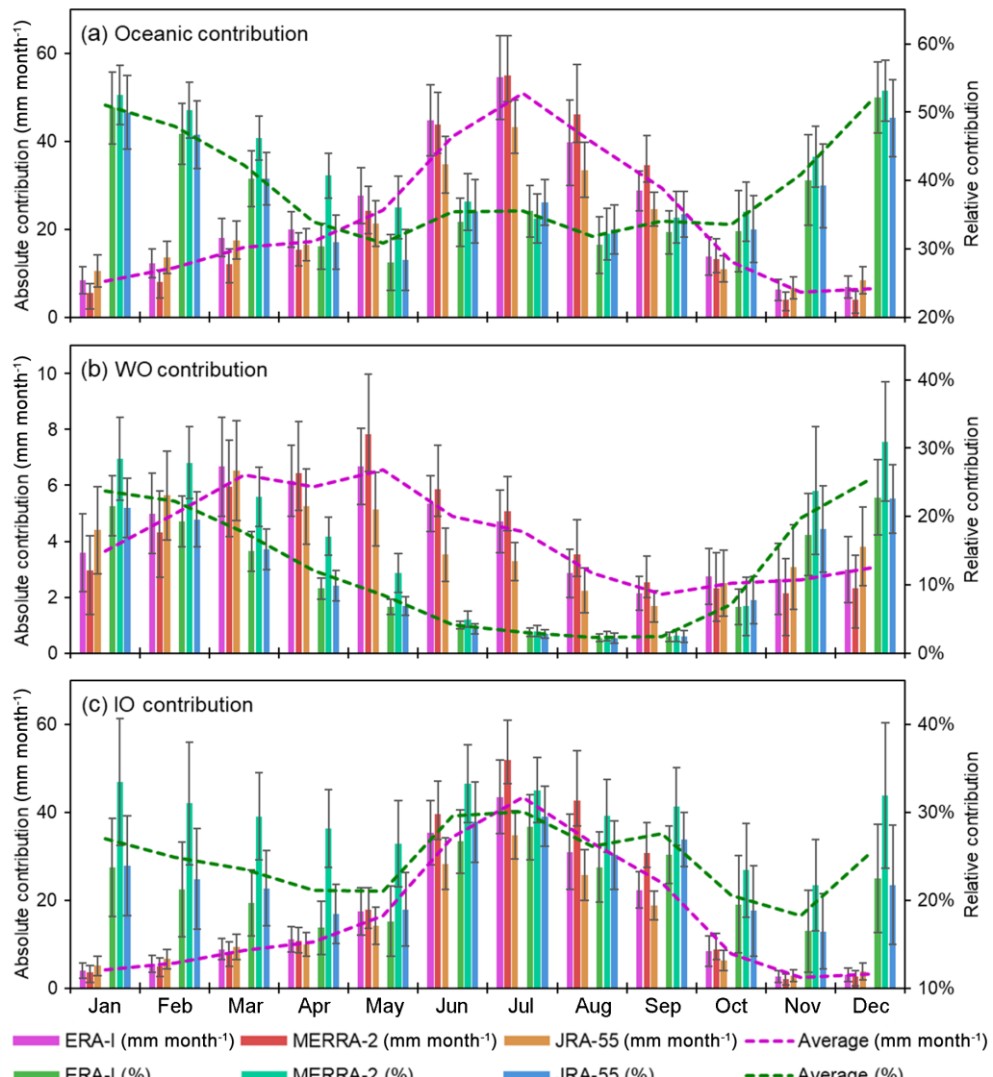

**Figure 2:** Intra-annual variations of long-term mean absolute and relative oceanic moisture contributions to the TP precipitation. (**a**), (**b**), and (**c**) represent the moisture contributions from the global oceans, the western oceans (WO), and the Indian Ocean (IO), respectively. Pink, red, and yellow bars are absolute contributions; and green, cyan, and blue bars are relative contributions. All oceanic contributions are simulated using WAM-2layers driven by ERA-Interim (1979–2015), MERRA-2 (1980–2015), and JRA-55 (1979–2015). Dashed pink and green lines are absolute and relative contributions, respectively, averaged across three simulations with different reanalysis datasets. Error bars represent one standard deviation of the interannual variations.

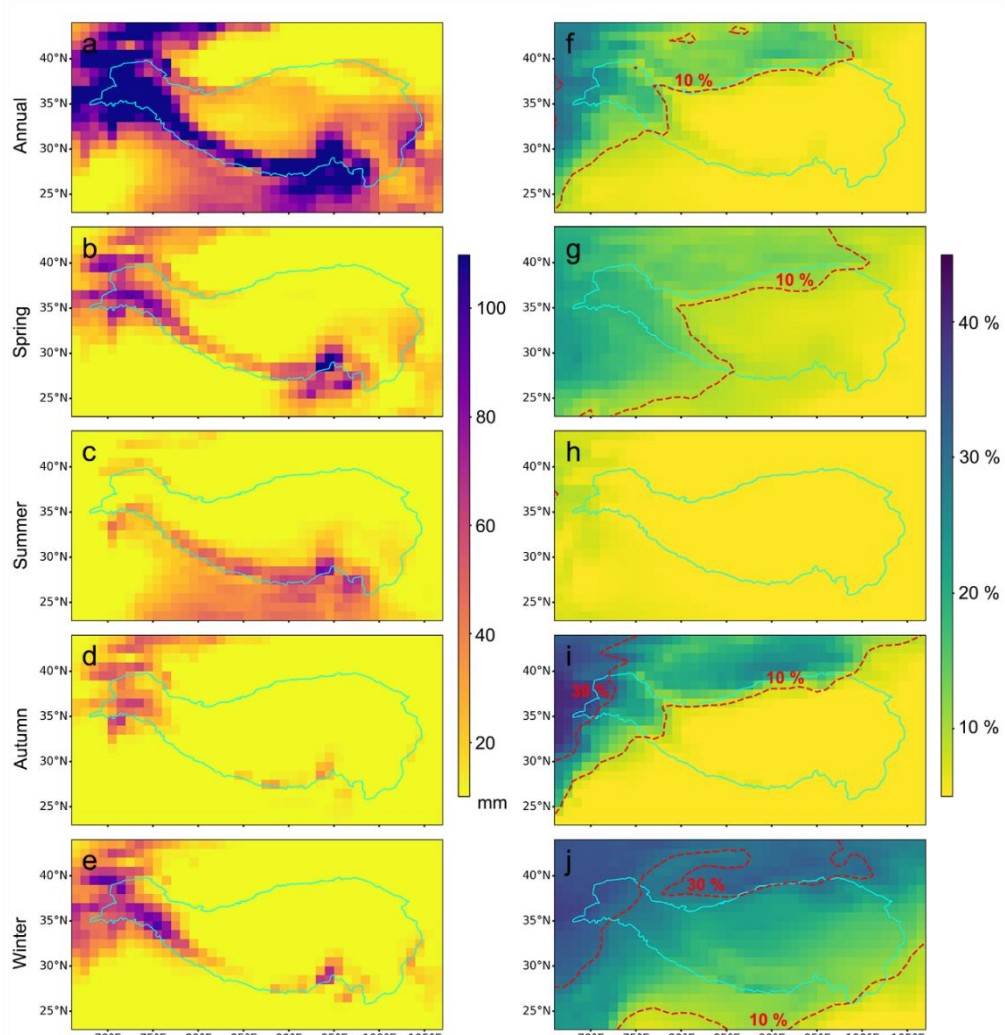

**Figure 3:** Spatial distributions of long-term mean absolute and relative moisture contribution from the western oceans (WO) to the TP precipitation. (**a**–**e**) The absolute contribution from the WO on annual (mm year$^{-1}$) and seasonal (mm season$^{-1}$) scales. (**f**–**j**) The relative contribution of the WO moisture on annual and seasonal scales. The dashed red lines in (**f**–**j**) are 10% and 30% isolines of the relative contribution. The forward moisture tracking results are modelled using WAM-2layers driven by ERA-Interim during 1979–2015. Moisture-tracking results driven by MERRA-2 (1980–2015) and JRA-55 (1979–2015) are shown in Figures S7 and S8, respectively.

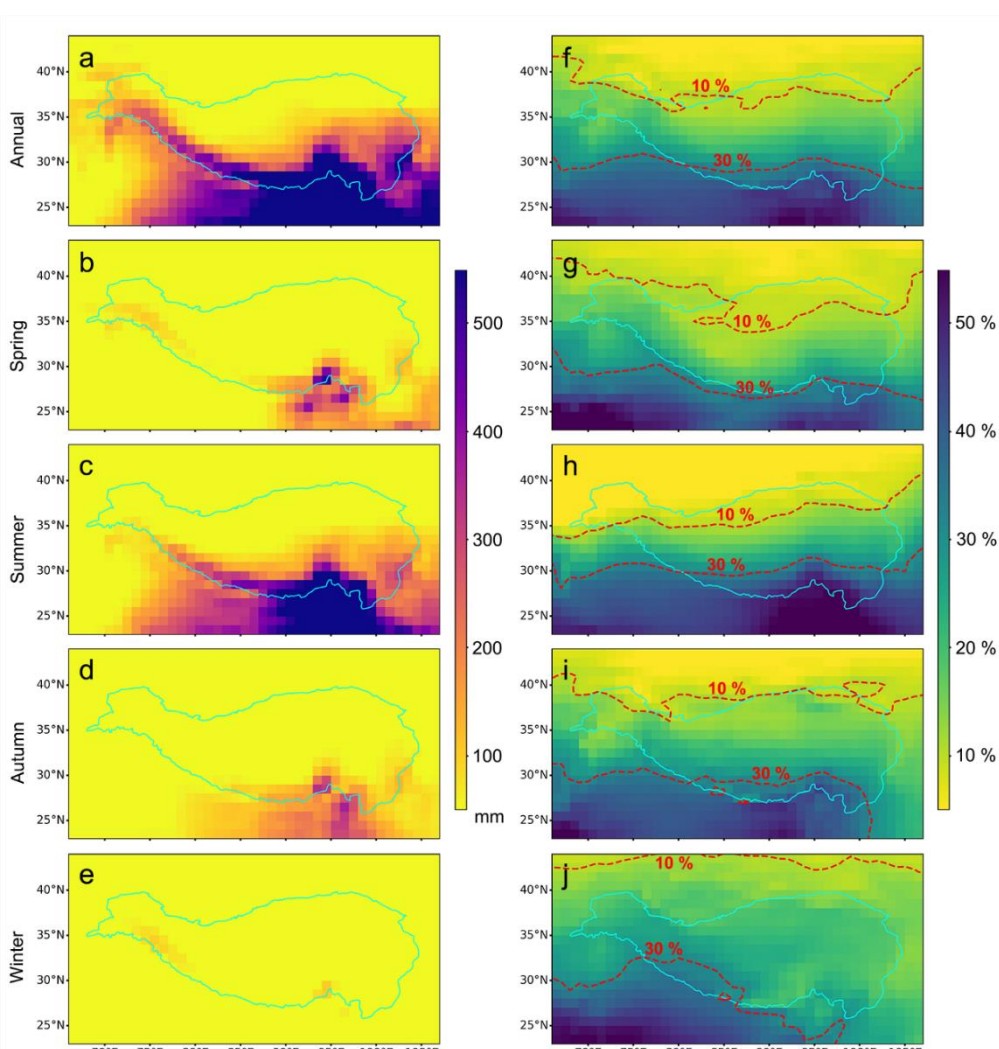

**Figure 4:** Spatial distributions of long-term mean absolute and relative moisture contribution from the Indian Ocean (IO) to the TP precipitation. (**a**–**e**) The absolute contribution from the IO on annual (mm year$^{-1}$) and seasonal (mm season$^{-1}$) scales. (**f**–**j**) The relative contribution of the IO moisture on annual and seasonal scales. The dashed red lines in (**f**–**j**) are 10% and 30% isolines of the relative contribution. The forward moisture tracking results are modelled by WAM-2layers forced with ERA-Interim during 1979–2015. Moisture-tracking results driven by MERRA-2 (1980–2015) and JRA-55 (1979–2015) are shown in Figures S9 and S10, respectively.

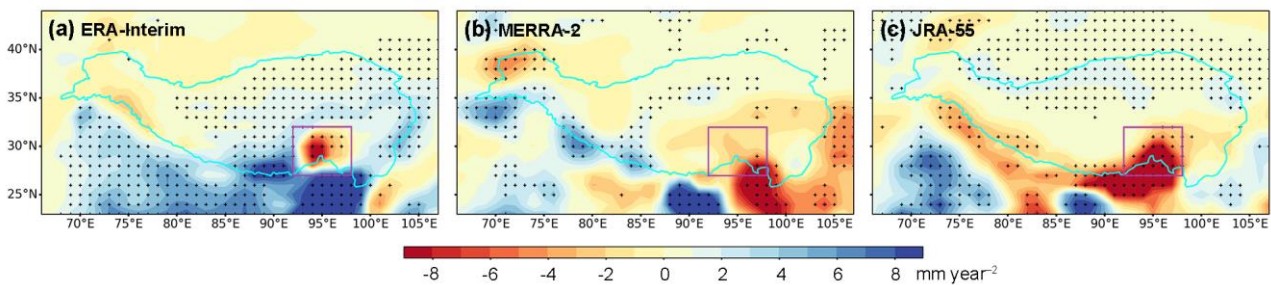

**Figure 5:** Long-term trends of oceanic moisture contribution to the precipitation over the TP region on the annual scale using (**a**) ERA-Interim (1979–2015), (**b**) MERRA-2 (1980–2015), and (**c**) JRA-55 (1979–2015). Stippling indicates regions with statistically significant trends ($p < 0.05$). The purple rectangle represents the SETP.

550

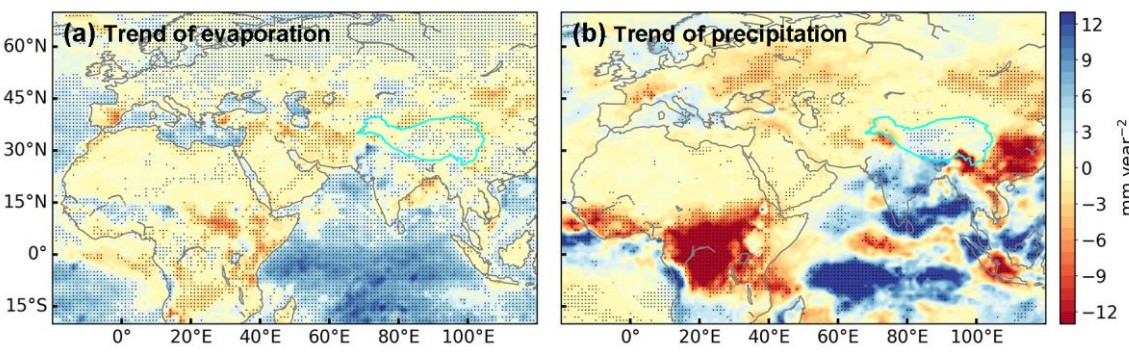

**Figure 6:** Long-term trends of annual (**a**) evaporation and (**b**) precipitation over possible source regions during 1979–2015. Stippling indicates regions with statistically significant trends ($p < 0.05$)

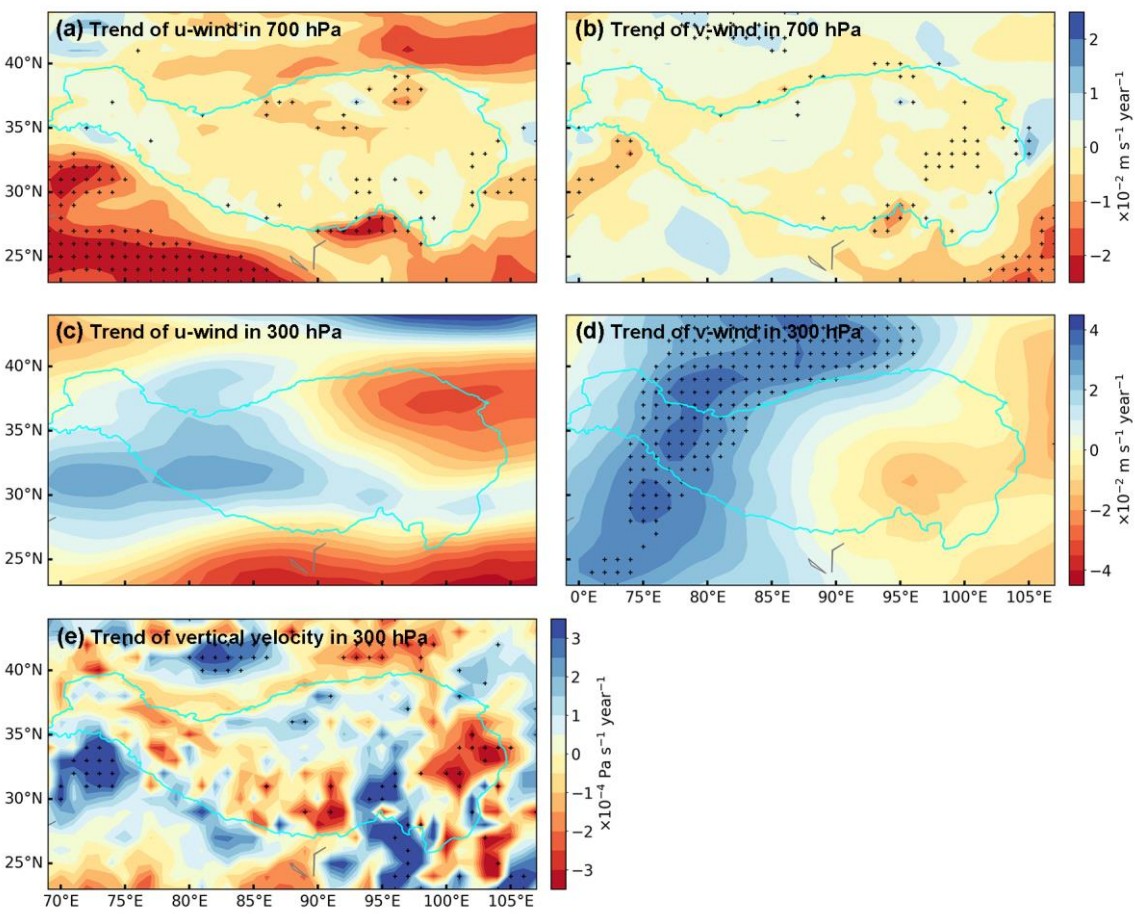

**Figure 7:** Long-term trends of annual (**a**) zonal (u) wind at 700 hPa (positive denotes enhanced eastward wind), (**b**) meridional (v) wind at 700 hPa (positive denotes enhanced northward wind), (**c**) zonal (u) wind at 300 hPa, (**d**) meridional (v) wind at 300 hPa, and (**e**) vertical velocity at 300 hPa (positive denotes decreased upward motion) around the TP region during 1979–2015. Stippling indicates regions with statistically significant trends ($p < 0.05$). Note the negative values of vertical velocity indicate upward motion.

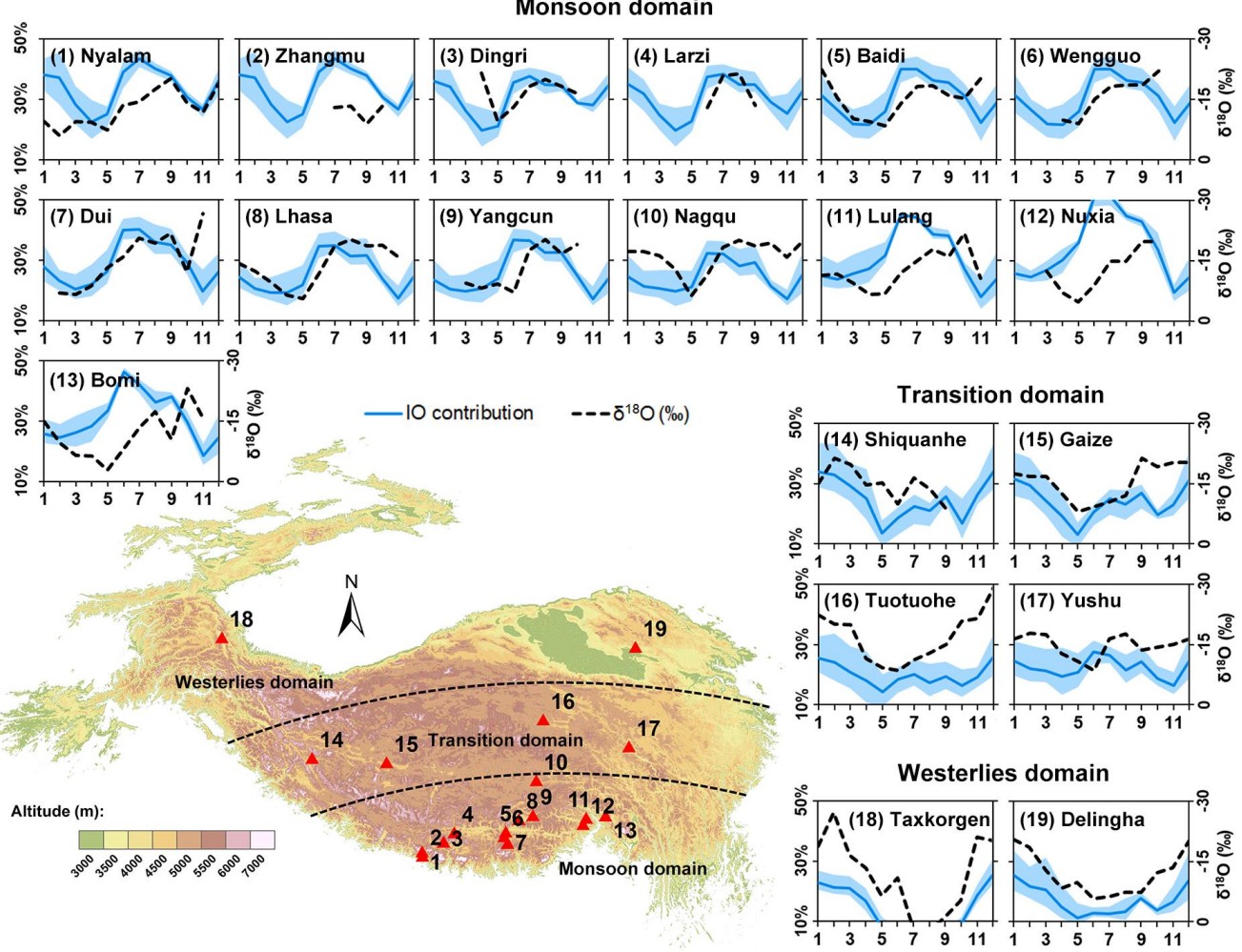

**Figure 8:** Locations of the precipitation isotope monitoring stations and the relationship between the monthly relative IO moisture contributions (blue lines) and the precipitation isotope observations (dotted lines). Sites 1–13, 14–17, and 18–19 represent stations located within the monsoon domain, transition domain, and westerlies domain, respectively. Blue lines show the mean IO moisture contributions based on three simulations, while the shadings show the range (detailed seasonal variations of three simulations are shown in Figure S14). Note that for consistency, oceanic contributions below 10% and above 50% are not shown for sites 12 and 18.