# Peer review of "Spatial Distribution of Oceanic Moisture Contributions to Precipitation over the Tibetan Plateau"

_Hydrology and Earth System Sciences, 2022_

## Referee Comment (RC1)

Review of the manuscript "Quantify Oceanic Moisture Contribution to the Tibetan Plateau " by Ying Li et al.

Despite the moisture sources of the Tibetan Plateau (TP) have been basically revealed by several researches on the basis of different methods, the further study of the moisture sources over TP from various dataset is necessary. In this manuscript, the authors quantified the absolute and relative contributions of oceanic moisture sources over TP based on a moisture tracking model and the various atmospheric reanalysis products. The methods in the manuscript is generally effective, while the moisture tracking method in this study still have non-ignorable uncertainties and need proper evaluation. In science, the novel contribution of this study is not clear due to most of the conclusions have been revealed in previous studies. Therefore, I would recommend that the manuscript need major revision before accepted by HESS. Below are my specific comments.

Specific comments:

1. In this study, there are several approximation in the Eulerian moisture tracking method (Van der Ent et al., 2010; Van der Ent, 2014), which induce non-ignorable uncertainties of the moisture sources calculations. For example, it only can resolve two vertical layers in the model and does not consider all the water substances (water vapour, cloud droplets, cloud ice, rain, and snow) and all the physical processes that the moisture undergo in the model, eg. deep convection, shallow convection, cloud macrophysics, cloud microphysics, diffusion etc. It is not the best one. In fact, the detailed quantified moisture models have been developed. In the references in around line 70, I suggest the authors pertinently evaluate previous studies and properly evaluate Van der Ent' (2014) method in section 2.1. Also need indicate the uncertainties of this method in the manuscript.

2. In science, the novel contribution of this study is not clear. The absolute and relative contributions of moisture sources, including oceanic source over TP have been quantitatively revealed. I suggest the authors focus on the comparisons of moisture sources evaluation based on the various atmospheric reanalysis products. The relationship between model oceanic source and isotope δ18O is interesting.

3. In line 147, I do not think the oceanic sources of the Mediterranean, the Red Sea, and the Persian Gulf, can compared to the Atlantic. They are too small. If say this, please give the quantitative tracking results.

4. Please indicate the sub-figures when describe in around line 239.

---

## Referee Comment (RC2)

[referee-annotated manuscript omitted]

---

## Author Comment (AC2)

**Response to Referee #2 (Ruud van der Ent):**

**General comments:** The authors analyzed the moisture sources of the Tibetan Plateau using 3 different reanalysis products, a widely used moisture tracking method WAM2layers and additional stable isotope data.

English editing of the paper is absolutely necessary as there are many small mistakes, but this can easily be solved using and English editing service.

Scientifically, the paper is clear, but the whole analysis can also be considered rather straightforward meaning that the novelty is somewhat minor. Obviously not all papers have to be major breakthroughs, but it would be nice if the authors could indicate a bit more specific what we now know that we did not know before from other studies that analyzed the moisture sources of the Tibetan Plateau.

My major comment regards the analysis in subsection 3.3 and the conclusion that a decrease in oceanic moisture contribution resulted in reduced TP precipitation. I have strong reservations with this conclusion since as far as I can see the cause and effect could very well be the other way around. This needs more detailed investigation and possibly less strong conclusions.

I attach more specific comments as a supplement.

**Response:** We are very grateful for your thorough review and comments, which help improve our manuscript and provide guidance for our future research.

**1.** We regret that there were problems with English writing. The paper will be carefully revised by all authors to improve the grammar and readability. The English editing service will be considered in our revision.

**2.** For the novelty of this work, in the revision, we will more clearly point out the unique contributions of this paper by (1) summarizing knowledge gaps and (2) thoroughly comparing this study with existing moisture tracking studies in the TP (see Table AC1). Please see our detailed response to specific comment #5.

**3.** For the analyses of oceanic moisture contribution and precipitation change over the TP in Section 3.3, we will thoroughly revise this section according to your specific comments #15, #16, #19, and #22. Please see our detailed response to specific comment #15.

Please see our responses to your specific comments below.

**Specific comments:**

1. In line 1 (*Title*).
**Comments:** In this way the title is 'imperative' which probably is not really intended.
**Response:** Thanks for pointing this out. Considering both reviewers' comments, we will change the title to "Spatial Distribution of Oceanic Moisture Contributions to the Tibetan Plateau". This title emphasizes the 'Spatial Distribution' which is the core novelty in comparison with previous studies in the TP (please see our detailed response to specific comment #5).

2. In line 11 (*Although recent accelerated global hydrological cycle*).

**Comments:** Incorrect English: probably meant: 'although the global hydrological cycle recently accelerated'. Contentwise: what does recently exactly mean?

**Response:** Thanks for pointing this out. We will correct grammatically issues in our revision. Specifically, this sentence will be revised to "Although the accelerated global hydrological cycle, the altered sea-land thermal contrast, and the amplified warming rate over the TP during the past several decades are known to have profound effects on the regional water balance, the contribution of oceanic evaporation, in particular its spatial variability over the vast TP, remains unclear."

3. In lines 25–26 (*Van der Ent et al., 2010; Trenberth et al., 2011*).
**Comments:** Aren't there more recent estimates?

**Response:** Thanks for reminding. For the description 'evaporation from oceans constitutes more than 80% of the global surface evaporation', the value '80%' was from the *Trenberth et al.* (2011) which quantified the global oceanic evaporation and terrestrial evapotranspiration using six reanalysis products (NCEP-NCAR R1/R2, CFSR, C20r, ERA-40, ERA-Interim, JRA-25, and MERRA). We considered this estimate robust and trustworthy.

For the description 'evaporation from oceans contributes to about 60% of terrestrial precipitation', the value '60%' was from *Van der Ent et al.* (2010) (one of the earliest studies). Later on, this estimate was mentioned by *Van der Ent and Savenije* (2013). More recently, *Link et al.* (2020) released a dataset on the fate of land evaporation where the information on the sources of precipitation can be extracted. In the revision, we will add these two extra sources (*Van der Ent and Savenije* (2013) and *Link et al.* (2020)) here.

4. In line 28.
**Comments:** Insert "the".
**Response:** Thanks. We will correct this in the revision.

5. In lines 56–57 (*However, the spatial variation of the oceanic moisture contribution from the Himalayas to the inner TP and their historical changes have not been examined yet*).
**Comments:** It's unclear to me why the authors specifically highlight the oceanic moisture contribution here. In principle there have been several recent studies about the moisture sources of the TP (including the oceanic ones) that have been overlooked here:
Guo, L., van der Ent, R. J., Klingaman, N. P., Demory, M.-E., Vidale, P. L., Turner, A. G., Stephan, C. C., and Chevuturi, A.: Moisture Sources for East Asian Precipitation: Mean Seasonal Cycle and Interannual Variability, 20, 657–672, https://doi.org/10.1175/JHM-D-18-0188.1, 2019.
Zhang, C., Tang, Q., Chen, D., van der Ent, R. J., Liu, X., Li, W., and Haile, G. G.: Moisture Source Changes Contributed to Different Precipitation Changes over the Northern and Southern Tibetan Plateau, J. Hydrometeor., 20, 217–229, https://doi.org/10.1175/JHM-D-18-0094.1, 2019.
Not only would a citation to these works be appropriate, but also: 1) How does this

paper add to what we already know from the aforementioned papers?    2) How do the results from this paper compared to the findings of the aforementioned papers?

**Response:** Thanks for the comments. We would like to address your concern from the following two aspects.

**(1) The pressing need to study the oceanic moisture contribution to the TP?**

Firstly, the TP has been considered as a thermal "air pump" that attracts low-latitude oceanic evaporation to the region, particularly under recently altered land-sea thermal gradient between the TP and global oceans (meteorological records revealed that the atmospheric warming rate over the TP was twice that of the global mean). A quantitative, spatial and temporal evaluation of the oceanic moisture contribution to the TP could help better understand the changing hydrological cycle over the TP and its underlying mechanisms.

Secondly, the interpretations of paleoclimate records in the TP, particularly the $\delta^{18}O$ and $\delta D$ in the precipitation and ice-cores, largely rely on the understanding of different moisture sources for the TP. For example, the $\delta^{18}O$ and $\delta D$ evaporated from oceans are relatively enriched in comparison with the other sources. Different oceanic contributions may link to different isotope values in different climate regions of the TP, which has not been thoroughly explored.

More specifically, we distinguished the moisture contribution of the Indian Ocean (IO) from that of the Western Oceans (WO) in our analyses. These two regions represent the source areas of the Indian summer monsoon and the mid-latitude westerlies (the two core circulation systems dominate the TP's climate), respectively. For example, by using numerous $\delta^{18}O$ measurements from precipitation and ice-core on the TP, Tian et al. (2007), Yao et al. (2013), and numerous isotope-related studies (Tian et al., 2001; Yu et al., 2008; Hren et al., 2009; Zhao et al., 2012; Joswiak et al., 2013; Ren et al., 2021) empirically identified a line around the 34°–35°N to represent the northward extension of the Indian summer monsoon. In this context, we intend to provide a quantitative view of the region influenced by the Indian monsoon, from the perspective of moisture contributions.

**(2) The novelty of this study as compared with previous moisture tracking studies in the TP.**

In comparison with the traditional synoptic and climatological analyses, the numerical moisture tracking method could quantitatively diagnose the moisture contribution to a target region. In Table AC1 below, we summarize existing studies using numerical moisture tracking in the TP published during the past two decades. Although these studies have quantified the oceanic moisture contribution to different parts of the TP in different seasons after the 1960s, nearly all of them only considered *regional averages* for specific target areas in the TP ('Study area' in Table AC1) with *backward* moisture tracking. The *spatial distribution* of oceanic moisture contribution to the vast TP, e.g., the transition gradient of the moisture transported from the Indian Ocean, is hitherto unclear. To fill this knowledge gap, in this study we leveraged a *forward* moisture tracking method and studied the *spatial distribution* of oceanic moisture contribution over the TP.

**Table AC1.** Studies of numerical moisture tracking in the TP region.

| Reference | Study area | Time period | Model | Data | Main conclusions |
|---|---|---|---|---|---|
| Chen et al. (2012) | TP | 2005–2009 (summer) | FLEXPART | NCEP/GFS | The ocean source could extend from the Arabian Sea to the Southern Hemisphere. |
| Sun and Wang (2014) | Grassland on eastern TP | 2000–2009 | FLEXPART | NCEP-CFSR | During the warm (cold) season, oceanic moisture is mainly from the Arabian Sea and Bay of Bengal (areas surrounding the Arabian Peninsula). |
| Zhang et al. (2017) | Central-western TP | 1979–2013 | WAM | ERA-I, NCEP-2 | More than 21% of the moisture comes from oceans. |
| Huang et al. (2018) | Southeastern TP | 1979–2016 (winter extreme precipitation) | LAGRANTO | ERA-I | About 18% of the moisture comes from oceans. |
| Pan et al. (2018) | Southern/northern TP | 1982–2014 | CAM | MERRA | During summer, the Indian Ocean supplies about 28.5% of the moisture to the southern TP. |
| Chen et al. (2019) | Four areas in TP | 1980–2016 (May–August) | FLEXPART | ERA-I | The northwestern TP and northeastern TP are less affected by the Indian monsoon moisture. |
| Guo et al. (2019) | TP | 1979–2015 | WAM-2layers | ERA-I | Indian Ocean and Pacific Ocean account for 24% and 2% of the moisture contribution, respectively. |
| Li et al. (2019) | Endorheic TP | 1979–2015 | WAM-2layers | ERA-I, MERRA-2, JRA-55 | 24%–30% of the moisture comes from oceans. |
| Qiu et al. (2019) | Three areas in TP | 1979–2016 (winter extreme precipitation) | LAGRANTO | ERA-I | Moisture contributions of the Arabian Sea to the intense precipitation in the western, south-central, and southeastern TP are 9.2%, 6.9%, and 1.1%, respectively. |
| Xu and Gao (2019) | Southeastern TP | 1982–2011 (April–September) | QIBT | ERA-I | Only 2% of the moisture originates from the oceanic source. |
| Zhang et al. (2019a) | Southern/northern TP | 1979–2016 | WAM-2layers | ERA-I | Northwestern (southeastern) source contributes ~39% (~51%) of the moisture in the northern (southern) TP. |
| Zhang et al. (2019b) | Sanjiangyuan Region | 1960–2017 (June–September) | HYSPLIT, HDBSCAN | NNR1 | About 51% (54%) of the medium to heavy precipitation is influenced by the northwestern (southern) source. |
| Liu et al. (2020) | Western TP | 1979–2018 (winter) | HYSPLIT | ERA-I | About 57% of the moisture comes from the Arabian Sea, Arabian Peninsula, and northern Indian Ocean. |
| Ma et al. (2020) | Seven areas in TP | 1961–2015 (summer extreme event) | HYSPLIT | NCEP/NCAR | About 75% of the moisture for extreme precipitation in the southeastern TP comes from the Bay of Bengal. |
| Yang et al. (2020) | Southeastern TP | 1980–2016 (June–September) | FLEXPART | ERA-I | 30% of the moisture comes from oceans. |
| Zhang (2020) | TP | 1998–2018 | WAM-2layers | ERA-I, TRMM | The southeastern source from the TP to the western Indian Ocean accounts for 32% of the moisture contribution. |

| LI et al. (2022) | Seven basins in TP | 1979–2015 | WAM-2layers | ERA-I, MERRA-2, JRA-55 | Oceanic moisture accounts for 24%–30% of the moisture in different basins of the TP. |

**6. In line 72.**

**Comments:** Insert "the".

**Response:** Thanks. We will correct this in the revision.

**7. In lines 82–84 (*and in comparison with Lagrangian models (e.g., FLEXible PARTicle (FLEXPART) dispersion model and the Hybrid SingleParticle Lagrangian Integrated Trajectory (HYSPLIT) model), the Eulerian grids enable the model to excel in computation speed and to consider moisture budget from precipitation and evaporation separately(Van der Ent et al., 2013; Van der Ent, 2014)*).**

**Comments:** This depends very much on what types of tracking runs are being done. Without going in depth in investigating this I would remove these claims entirely.

**Response:** Thank you for the suggestion. We will remove this inappropriate statement in the revision.

**8. In line 86 (*Equation 1*).**

**Comments:** There was an sign error in Van der Ent, 2014, better to write the equations as in Findell et al. (2019)

Findell, K. L., Keys, P. W., van der Ent, R. J., Lintner, B. R., Berg, A., and Krasting, J. P.: Rising Temperatures Increase Importance of Oceanic Evaporation as a Source for Continental Precipitation, J. Clim., 32, 7713–7726, https://doi.org/10.1175/JCLI-D-19-0145.1, 2019.

**Response:** Thanks for your reminder. The Equation (1) will be corrected to $\frac{\partial S_{g,lower}}{\partial t} = -\frac{\partial (S_{g,lower}u)}{\partial x} - \frac{\partial (S_{g,lower}v)}{\partial y} + E_g - P_g \pm F_{v,g}$ for forward moisture tracking in WAM-2layers in the lower layer. Accordingly, all the text description about the Equation will be revised in our further revision.

**9. In lines 91–92 (*Due the existence of residual $\xi_k$, the closure of the model is defined by a ratio of residuals between the two layers, i.e., $\xi_{top}/S_{top} = \xi_{bottom}/S_{bottom}$*).**

**Comments:** That is not the definition of closure, but an assumption that is used in order to calculate the vertical flux (see: van der Ent et al., (2014), appendix B).

van der Ent, R. J., Wang-Erlandsson, L., Keys, P. W., and Savenije, H. H. G.: Contrasting roles of interception and transpiration in the hydrological cycle - Part 2: Moisture recycling, 5, 471–489, https://doi.org/10.5194/esd-5-471-2014, 2014.

**Response:** Thanks for pointing this out. We will thoroughly inspect all the incorrect or improper descriptions in the method section. Due to the modification of the water balance equation in forward moisture tracking (Equation 1), and according to the relevant descriptions in Van der Ent et al. (2013), van der Ent et al. (2014), and Findell et al. (2019), we revised this part to: "The 'well-mixed' assumption is applied to this model, which means the precipitation is assumed to be immediately removed from the

atmosphere in the tracking process (i.e., $P_g/P = S_g/S$, where subscript $g$ denotes the targeted moisture, and $P$ and $S$ are total precipitation and total column atmospheric moisture storage, respectively). To better capture the vertical exchanges due to convection, turbulence, and re-evaporation and minimize the water balance losses between the two layers, the gross vertical flow is set to 4 times the vertical flow in the net flow direction and 3 times the vertical flow in the opposite direction. Although this is a simplification of the turbulent moisture exchange, physically reasonable results have been obtained, and the general tracking has been validated against an online 3D tracking model (Van der Ent et al., 2013; van der Ent et al., 2014; Findell et al., 2019)."

10. In line 94 (*1°×1°, and the time step is set as 0.25 h*).
**Comments:** This is at a higher resolution than Van der Ent et al. (2014), who used a 1.5 arcdegree resolution. Yet, the authors have chosen the same time step. This may lead to instable and spurious results at high latitudes or at least internal model corrections to maintain stability.
**Response:** Thanks for this comment. We have tested the sensitivity of our results to the selection of different time steps. Below is an example, which compares the results using a 15-min time step (0.25 h) and a 10-min time step. As suggested in Figure AC1a and b, visually the results of mean annual oceanic moisture contribution to the TP with different time steps are nearly identical. This is also confirmed by the differences of these two runs, as shown in Figure AC1c and d. Discrepancies in moisture tracking results induced by different time step mainly appear in the western TP (Figure AC1c), although very minor (~1 mm on annual scale). The relative differences are below 1% in the TP on annual scale (Figure AC1d). This suggests the stability of using different time steps in the study area.

[Figure]

**Figure AC1.** Simulations of mean annual oceanic moisture contribution to the TP with 0.25-h (15-min) time step (a) and 10-min time step (b). (c) is the difference (mm year$^{-1}$) between the two simulations. (d) is the relative difference (% year$^{-1}$) between the two simulations.

Please note that as limited by our response deadline, we only performed annual-scale

sensitivity analysis with ERA-I at the current stage. Other comparisons will be done in our revision. In our revised manuscript, we will add a part to evaluate the potential numerical instability that may be triggered by different time-steps.

11. In line 95 (*as around 812 hPa*).
**Comments:** But varying with surface pressure?! This is a very important detail.
**Response:** We will revise this sentence as: "The vertical separation between the two layers is prescribed as around 812 hPa at the normal atmospheric pressure. Note that the atmospheric pressure of the vertical separation varies with surface pressure, e.g., the "half-level" pressure in the model is defined as $P_{k-1/2} = A_{K-1/2} + B_{k-1/2}P_s$ where $P_s$ is surface pressure, $k$ represents different model levels in different reanalysis products, and the values of $A_{K-1/2}$ and $B_{k-1/2}$ are independently defined by different reanalysis products.

12. In lines 102–103 (*The ocean and land distributions were defined according to the 1°×1° gridded land-sea mask from ERA-Interim*).
**Comments:** The land-sea mask of ERA-Interim considers lakes to be 'sea', but it does not makes sense to consider them 'ocean' in this tracking study in my opinion.
**Response:** Thanks for pointing this out. In our simulation, we removed all inland large lakes (considered 'sea' in land-sea mask in ERA-I), for example, the Caspian Sea and the Black Sea. The final land-sea mask with the 1°×1° spatial resolution used in this work is shown in Figure AC2. In addition, our intention to use this land-sea mask was that it is more suitable for the precipitation isotope studies over the TP. In the revision, we will add Figure AC2 in Supplementary.

[Figure]

**Figure AC2.** The land-sea mask used in our manuscript with 1°×1° spatial resolution (the red covered area represents the ocean area).

13. In line 109 (*6h, 17 layers*).
**Comments:** What kind of layers?
**Response:** Table AC2 summarizes model layers in the three reanalysis products we used in this study. This table will be added to the Supplementary in our revision.

**Table AC2.** The selected 17 model layers in three reanalysis products.

|  | Model layers (from surface to upper atmosphere) | | |
|  | ERA-Interim | MERRA2 | JRA55 |
| --- | --- | --- | --- |
| 1 | 60 | 72 | 1 |
| 2 | 59 | 71 | 2 |
| 3 | 58 | 70 | 3 |
| 4 | 57 | 69 | 4 |
| 5 | 56 | 68 | 5 |
| 6 | 55 | 67 | 6 |
| 7 | 54 | 66 | 7 |
| 8 | 51 | 65 | 9 |
| 9 | 48 | 61 | 12 |
| 10 | 47 | 59 | 14 |
| 11 | 44 | 55 | 17 |
| 12 | 41 | 52 | 20 |
| 13 | 38 | 49 | 23 |
| 14 | 35 | 46 | 26 |
| 15 | 32 | 44 | 29 |
| 16 | 27 | 40 | 34 |
| 17 | 17 | 28 | 44 |

14. In line 123 (*in mm and %*).

**Comments:** mm per what? precipitation is a flux with dimension of length x Time-1, where x is 1,2 or 3; percent of what? of the total local evaporation or of the total sink precipitation or something else?

**Response:** Sorry for the confusion. All these ambiguous units will be corrected in our revision. In Figures 1, 3, and 4, "mm" will be corrected to "mm year$^{-1}$". In Figure 2, "mm" will be corrected to "mm month$^{-1}$". In Figure 5, "mm yr$^{-1}$" will be corrected to "mm year$^{-2}$". Figures in the Supplementary will also be revised accordingly. The unit % represents the percentage of the total sink precipitation, which will be mentioned in the revised method Section.

15. In line 187 (*Section 3.3*).

**Comments:** This seems to be one of the core results yet cause and effect could be entirely reversed, meaning precipitation on TP declines and as a consequence the oceanic contribution also drops, possibly keeping exactly the same ratio. However, it may also be that the evaporation of the ocean has dropped or that the source area has changed. Simply looking at similarities in Figure 5 is insufficient proof in my opinion. Moreover, this subsection discusses many results in the supplement, but if they are discussed at length they should be in the main text instead.

**Response:** Per your comments, we performed additional analyses and concluded that the decreased oceanic contribution is mainly induced by precipitation decrease over TP. We will thoroughly revise the Section 3.3 to improve the clarity following the structure below:

  **a.** Analyze the long-term trends of oceanic moisture contribution to the TP (Figure

AC3), and raise the question that decreased oceanic moisture contribution was found mainly around the southeastern TP (i.e., the Brahmaputra Canyon region, which has long been considered the most important moisture transport channel of the TP (*Hren et al.*, 2009)). More specifically, this decreased oceanic moisture contribution was found to originate from both the monsoon-dominated Indian Ocean (IO) and the westerlies-dominated Western Oceans (WO) (Figure AC4).

[Figure]

**Figure AC3.** Trends of oceanic moisture contribution to the TP region with ERA-Interim (1979–2015), MERRA-2 (1980–2015), and JRA-55 (1979–2015).

[Figure]

**Figure AC4.** Trends of oceanic moisture contribution to the TP region from the Indian Ocean (IO, a–c) and the western oceans (WO, d–f) with ERA-Interim (1979–2015), MERRA-2 (1980–2015), and JRA-55 (1979–2015).

**b.** Explain the trends of oceanic moisture contribution by detecting the changes of oceanic evaporation, precipitation, the horizontal wind fields, and the updraft around the TP region. According to the reviewer's suggestion, we calculated the inter-annual trends of global evaporation and precipitation during 1979–2015 (Figure AC5). Most of the oceanic sources exhibit *enhanced* evaporation. However, the moisture may lose significantly due to precipitation along the transport pathway, particularly when the moisture transport across the Indian Subcontinent (Figure AC5b). In addition, the inter-annual trends of zonal (u) and meridional (v) wind in 700 hPa and 300 hPa and vertical velocity in 300 hPa are also analyzed (Figure AC6). Significantly weakened eastward and northward winds in the lower atmosphere (700 hPa) are found around the southeastern TP (the changes of horizontal wind fields in the 300 hPa are not significant in the region). This may indicate decreased moisture convergence in the region from the lower atmospheric transport. At the same time, significantly decreased upward motion (negative values of vertical velocity indicate upward motion) in the higher

atmosphere (300 hPa, Figure AC6e) was found in the southeastern TP. This further verifies that less moisture is condensed as precipitation in this region.

[Figure]

**Figure AC5.** Trends of global evaporation (a), precipitation (b), during 1979–2015. Stippling indicates regions with statistically significant trends ($p < 0.05$)

[Figure]

**Figure AC6.** Trends of u-wind at 700 hPa (a), v-wind at 700 hPa (b), u-wind at 300 hPa (c), v-wind at 300 hPa (d), and vertical velocity at 300 hPa (e) around the TP region during 1979–2015. Stippling indicates regions with statistically significant trends ($p < 0.05$).

**c.** When studying the inter-annual trends of precipitation over the TP (Figure AC5b), a similar spatial pattern (significantly decreased precipitation over the southeastern TP) was found in comparison with the trends of oceanic moisture contribution to the TP region (Figure AC3). To detect whether the oceanic moisture contribution is connected to the spatial pattern of precipitation change over the TP, we carried out additional backward moisture tracking over the southeastern TP (Figure AC7a). The southeastern TP (SETP) was defined as the purple rectangle in Figure AC3 where the oceanic

moisture contribution and the precipitation both show a decreasing trend during 1979–2015. The spatial distribution of the trends of moisture source contributions to the SETP during the period is also shown in Figure AC7b.

As shown in Figure AC7b, the decreased moisture contributions to precipitation over the SETP are found for both the westerlies-dominated western sources and the monsoon-dominated southern sources. Meanwhile, only the southwestern slope of the Himalayas and the southwestern corner of the TP show increased moisture contribution to the SETP (Figure AC7b). Overall, the decreased oceanic moisture contribution does not dominate the precipitation change over the SETP, although they happen to have similar spatial patterns.

Per the reviewer's suggestion, we will move some figures that show key results from supplementary to the main text in our revision.

[Figure]

**Figure AC7.** (a) Long-term mean moisture source to precipitation in the SETP and (b) the relevant trends of moisture source contributions during 1979–2015, by using WAM-2layers forced with ERA-Interim. The blue rectangles represent the SETP. Stippling indicates regions with statistically significant trends ($p < 0.05$).

16. In line 212–213 (*Here we further reveal that this dipole pattern is driven by the changes in oceanic moisture contribution (particularly from IO)*).
**Comments:** As said before cause and effect may not be so easy to separate. Moreover, the authors should give an explanation of why the oceanic moisture contribution drops: which can be: less TP precipitation overall, less oceanic evaporation, changing pattern, ...
**Response:** Thanks for the comment. Based on our new analysis above, we concluded that decreased oceanic contribution is mainly induced by precipitation decrease over TP. This will be reflected in our thoroughly revised Section 3.3 in the revision. Please see our detailed response to comment #15 for details.

17. In line 222 (*The strongest relationship is found between precipitation δ18O and relative oceanic moisture contribution from IO (Figure 6)*).
**Comments:** These are interesting plots, but these relationships should be quantified with at least a correlation metric.
**Response:** Thanks. Per your comments, we calculated the correlation coefficients between the precipitation $\delta^{18}O$ and the relative oceanic moisture contribution from IO in the 19 stations on the TP (Table AC3). Note that some of the stations contain isotope

data less than 10 months (the short time-span may lead the correlation analyses less significance), thus, we only calculated the correlation coefficients with more than 10-month isotope data. In Table AC3, nearly all the precipitation $\delta^{18}O$ exhibit opposite correlations with the relative oceanic moisture contributions from IO, particularly in the westerlies-domain stations where all the correlations are significant ($p < 0.05$). These conclusions are consistent with our description in Section 3.4. We will add these quantitative results to Section 3.4 in the revision.

**Table AC3.** The correlation coefficients of seasonal changes in the 19 stations between the precipitation $\delta^{18}O$ and the relative oceanic moisture contribution from IO (derived from ERA-Interim, MERRA-2, and JRA-55, respectively). '*' represents statistically significant correlation coefficients ($p < 0.05$).

| | | Model layers (from surface to upper atmosphere) | | |
| --- | --- | --- | --- | --- |
| | | ERA-I | MERRA-2 | JRA-55 |
| Monsoon domain | 1.Nyalam | –0.65* | –0.18 | –0.51 |
| | 2.Zhangmu | - | - | - |
| | 3.Dingri | - | - | - |
| | 4.Larzi | - | - | - |
| | 5.Baidi | –0.37 | –0.42 | –0.41 |
| | 6.Wengguo | - | - | - |
| | 7.Dui | –0.38 | –0.49 | –0.33 |
| | 8.Lhasa | –0.62* | –0.44 | –0.52 |
| | 9.Yangcun | - | - | - |
| | 10.Nagqu | –0.39 | –0.18 | –0.05 |
| | 11.Lulang | –0.44 | –0.12 | –0.29 |
| | 12.Nuxia | - | - | - |
| | 13.Bomi | –0.05 | 0.30 | 0.16 |
| Transition domain | 14.Shiquanhe | - | - | - |
| | 15.Gaize | –0.73* | –0.52* | –0.36 |
| | 16.Tuotuohe | –0.80* | –0.63* | –0.36 |
| | 17.Yushu | –0.06* | 0.04 | 0.32 |
| Westerlies domain | 18.Taxkorgen | –0.87* | –0.87* | –0.84* |
| | 19.Delingha | –0.84* | –0.84* | –0.75* |

18. In line 228–231 (*Note the mismatches between summer peaks of relative moisture from IO and low $\delta^{18}O$ values in autumn at Lulang, Nuxia, and Bomi near the Brahmaputra Canyon, which is likely attributable to the impact of moisture transported from southeast Asia or the Pacific Ocean driven by the trough embedded in the southern branch of the westerlies(Cai and Tian, 2020)*).
**Comments:** These could be investigated in more detail with moisture tracking rather than simply relying on the Cai and Tian (2020) study.
**Response:** Thanks. Per your suggestion, we conducted additional moisture tracking for monthly moisture sources of the SETP (blue rectangle in Figure AC8, which contains Lulang, Nuxia, and Bomi stations) near the Brahmaputra Canyon in Figure AC8. From

June to September, moisture sources gradually extend to the southeast Asia and the western Pacific Ocean. This is in line with the finding in *Cai and Tian* (2020). We will add this figure in Supplementary in our revised manuscript.

[Figure]

**Figure AC8.** Mean monthly moisture source contributions to precipitation in the SETP, by using WAM-2layers forced with ERA-Interim (1979–2015). The blue rectangles represent the SETP.

19. In line 254 (*is more consistent with precipitation patterns*).
**Comments:** than what?
**Response:** Sorry for the confusion. We will revise this part based on the thoroughly revised Section 3.3. Please see our response to comment #15 above.

20. In line 266 (*Data availability*).
**Comments:** In my understanding the authors should here nowadays provide links to where their data (to reproduce their figures) can be found.

The availability of the forcing data can be described in methods, acknowledgements and/or references.

Code availability of the original and adapted WAM2layers model is entirely missing.
**Response:** Thanks for the comments. We plan to upload our raw data that can be used to reproduce Figures 1–5. Moreover, the datasets and code of WAM-2layers used in this work will be detailed in the references as following:

European Centre for Medium-Range Weather Forecast (ECMWF): The ERA-Interim reanalysis dataset, available at: https://apps.ecmwf.int/datasets/data/interim-full-daily/,

last access: 16 May 2017.

NASA Goddard Earth Sciences Data and Information Services Center (GES DISC): Modern-Era Retrospective analysis for Research and Applications, Version 2, available at: https://disc.gsfc.nasa.gov/datasets?project=MERRA-2, last access: 19 June 2018.

Japan Meteorological Agency: JRA-55: Japanese 55-year Reanalysis, Daily 3-Hourly and 6-Hourly Data, Archived at the National Center for Atmospheric Research, Computational and Information Systems Laboratory, available at: https://rda.ucar.edu/datasets/ds628.0/, last access: 19 July 2018.

Global Precipitation Climatology Centre (GPCC): GPCC Full Data Monthly Product Version 2018 at 1.0°: Monthly Land-Surface Precipitation from Rain-Gauges built on GTS-based and Historical Data, available at: https://www.dwd.de/EN/ourservices/gpcc/gpcc.html, last access: 26 August 2018.

National Tibetan Plateau Data Center: Data set of $\delta^{18}O$ stable Isotopes in Precipitation from Tibetan Network for Isotopes (1991–2008), available at: http://data.tpdc.ac.cn/en/, last access: 5 August 2022.

van der Ent, R. J. (15 July 2016): WAM-2layers v2.4.08, available at: https://github.com/ruudvdent/WAM2layersPython, last access: 5 August 2022.

21. In line 426 (*Figure 2*).
**Comments:** mm/month. if the x-axis represents month than write the abbreviations of the months instead of 1 – 12
**Response:** Thanks, we will correct this in our revised manuscript.

22. In line 447 (*Figure 5*).
**Comments:** It would be (more) relevant to look at trends in oceanic moisture contribution with respect to precipitation within the same reanalysis. I now think you're essentially comparing the different precipitation datasets, meaning these plots would have looked the same for total precipitation without any moisture tracking.
**Response:** Thanks. Per your comments #15, #16, #19, and #22, we will thoroughly revise Section 3.3, which also include the trends in oceanic moisture contribution and precipitation over the TP. Please see our detailed response to your comment #15 above.

**References:**

Cai, Z., and Tian, L.: What Causes the Postmonsoon 18O Depletion Over Bay of Bengal Head and Beyond?, Geophys. Res. Lett., 47(4), e2020GL086985, https://doi.org/:10.1029/2020GL086985, 2020.

Chen, B., Xu, X. D., Yang, S., and Zhang, W.: On the origin and destination of atmospheric moisture and air mass over the Tibetan Plateau, Theor. Appl. Climatol., 110(3), 423-435, https://doi.org/:10.1007/s00704-012-0641-y 2012.

Chen, B., Zhang, W., Yang, S., and Xu, X. D.: Identifying and contrasting the sources of the water vapor reaching the subregions of the Tibetan Plateau during the wet season, Climate Dyn., 53(11), 6891-6907, https://doi.org/:10.1007/s00382-019-04963-2, 2019.

Findell, K. L., Keys, P. W., van der Ent, R. J., Lintner, B. R., Berg, A., and Krasting, J. P.: Rising

Temperatures Increase Importance of Oceanic Evaporation as a Source for Continental Precipitation, J. Climate, 32(22), 7713-7726, https://doi.org/:10.1175/jcli-d-19-0145.1, 2019.

Guo, L., van der Ent, R. J., Klingaman, N. P., Demory, M.-E., Vidale, P. L., Turner, A. G., Stephan, C. C., and Chevuturi, A.: Moisture Sources for East Asian Precipitation: Mean Seasonal Cycle and Interannual Variability, J. Hydrometeorol., 20(4), 657-672, https://doi.org/:10.1175/jhm-d-18-0188.1, 2019.

Hren, M. T., Bookhagen, B., Blisniuk, P. M., Booth, A. L., and Chamberlain, C. P.: δ18O and δD of streamwaters across the Himalaya and Tibetan Plateau: Implications for moisture sources and paleoelevation reconstructions, Earth Planet. Sci. Lett., 288(1), 20-32, https://doi.org/:10.1016/j.epsl.2009.08.041, 2009.

Huang, W., Qiu, T., Yang, Z., Lin, D., Wright, J. S., Wang, B., and He, X.: On the formation mechanism for wintertime extreme precipitation events over the southeastern Tibetan Plateau, J. Geophys. Res.-Atmos., 123(22), 12,692-612,714, https://doi.org/:10.1029/2018JD028921 2018.

Joswiak, D. R., Yao, T., Wu, G., Tian, L., and Xu, B.: Ice-core evidence of westerly and monsoon moisture contributions in the central Tibetan Plateau, J. Glaciol., 59(213), 56-66, https://doi.org/:10.3189/2013JoG12J035 2013.

Li, Y., Su, F., Chen, D., and Tang, Q.: Atmospheric Water Transport to the Endorheic Tibetan Plateau and Its Effect on the Hydrological Status in the Region, J. Geophys. Res.-Atmos., 124(23), 12864-12881, https://doi.org/:10.1029/2019jd031297, 2019.

LI, Y., SU, F., TANG, Q., GAO, H., YAN, D., PENG, H., and XIAO, S.: Contributions of moisture sources to precipitation in the major drainage basins in the Tibetan Plateau, Sci. China-Earth Sci., 65(1674-7313), 1088, https://doi.org/:https://doi.org/10.1007/s11430-021-9890-6, 2022.

Link, A., van der Ent, R., Berger, M., Eisner, S., and Finkbeiner, M.: The fate of land evaporation - a global dataset, Earth System Science Data, 12(3), 1897-1912, https://doi.org/:10.5194/essd-12-1897-2020, 2020.

Liu, X., Liu, Y., Wang, X., and Wu, G.: Large-Scale Dynamics and Moisture Sources of the Precipitation Over the Western Tibetan Plateau in Boreal Winter, J. Geophys. Res.-Atmos., 125(9), e2019JD032133, https://doi.org/:10.1029/2019JD032133, 2020.

Ma, Y., Lu, M., Bracken, C., and Chen, H.: Spatially coherent clusters of summer precipitation extremes in the Tibetan Plateau: Where is the moisture from?, Atmos. Res., 237, 104841, https://doi.org/:10.1016/j.atmosres.2020.104841, 2020.

Pan, C., Zhu, B., Gao, J., Kang, H., and Zhu, T.: Quantitative identification of moisture sources over the Tibetan Plateau and the relationship between thermal forcing and moisture transport, Climate Dyn., 52(1-2), 181-196, https://doi.org/:10.1007/s00382-018-4130-6, 2018.

Qiu, T., Huang, W., Wright, J. S., Lin, Y., Lu, P., He, X., Yang, Z., Dong, W., Lu, H., and Wang, B.: Moisture Sources for Wintertime Intense Precipitation Events Over the Three Snowy Subregions of the Tibetan Plateau, J. Geophys. Res.-Atmos., 124(23), 12708-12725, https://doi.org/:10.1029/2019jd031110, 2019.

Ren, W., Tian, L., and Shao, L.: Regional moisture sources and Indian summer monsoon (ISM) moisture transport from simultaneous monitoring of precipitation isotopes on the southeastern and northeastern Tibetan Plateau, J. Hydrol., 601, 126836, https://doi.org/:10.1016/j.jhydrol.2021.126836, 2021.

Sun, B., and Wang, H.: Moisture sources of semiarid grassland in China using the Lagrangian particle model FLEXPART, J. Climate, 27(6), 2457-2474, https://doi.org/:10.1175/JCLI-D-13-

00517.1 2014.

Tian, L., Masson-Delmotte, V., Stievenard, M., Yao, T., and Jouzel, J.: Tibetan Plateau summer monsoon northward extent revealed by measurements of water stable isotopes, J. Geophys. Res.-Atmos., 106(D22), 28081-28088, https://doi.org/:10.1029/2001jd900186, 2001.

Tian, L., Yao, T., MacClune, K., White, J., Schilla, A., Vaughn, B., Vachon, R., and Ichiyanagi, K.: Stable isotopic variations in west China: a consideration of moisture sources, Journal of Geophysical Research: Atmospheres (1984–2012), 112(D10), D10112, https://doi.org/:10.1029/2006JD007718, 2007.

Trenberth, K. E., Fasullo, J. T., and Mackaro, J.: Atmospheric Moisture Transports from Ocean to Land and Global Energy Flows in Reanalyses, J. Climate, 24(18), 4907-4924, https://doi.org/:10.1175/2011JCLI4171.1, 2011.

Van der Ent, R. J., and Savenije, H. H.: Oceanic sources of continental precipitation and the correlation with sea surface temperature, Water Resour. Res., 49(7), 3993-4004, https://doi.org/:10.1002/wrcr.20296, 2013.

Van der Ent, R. J., Savenije, H. H., Schaefli, B., and Steele-Dunne, S. C.: Origin and fate of atmospheric moisture over continents, Water Resour. Res., 46(9), W09525, https://doi.org/:10.1029/2010WR009127, 2010.

Van der Ent, R. J., Tuinenburg, O. A., Knoche, H. R., Kunstmann, H., and Savenije, H. H. G.: Should we use a simple or complex model for moisture recycling and atmospheric moisture tracking?, Hydrol. Earth Syst. Sci., 17(12), 4869-4884, https://doi.org/:10.5194/hess-17-4869-2013, 2013.

van der Ent, R. J., Wang-Erlandsson, L., Keys, P. W., and Savenije, H. H. G.: Contrasting roles of interception and transpiration in the hydrological cycle – Part 2: Moisture recycling, Earth Syst. Dynam., 5(2), 471-489, https://doi.org/:10.5194/esd-5-471-2014, 2014.

Xu, Y., and Gao, Y.: Quantification of Evaporative Sources of Precipitation and Its Changes in the Southeastern Tibetan Plateau and Middle Yangtze River Basin, Atmosphere, 10(8), 428, https://doi.org/:10.3390/atmos10080428, 2019.

Yang, S., Zhang, W., Chen, B., Xu, X., and Zhao, R.: Remote moisture sources for 6-hour summer precipitation over the Southeastern Tibetan Plateau and its effects on precipitation intensity, Atmos. Res., 236, 104803, https://doi.org/:10.1016/j.atmosres.2019.104803, 2020.

Yao, T., Masson-Delmotte, V., Gao, J., Yu, W., Yang, X., Risi, C., Sturm, C., Werner, M., Zhao, H., and He, Y.: A review of climatic controls on δ18O in precipitation over the Tibetan Plateau: Observations and simulations, Rev. Geophys., 51(4), 525-548, https://doi.org/:10.1002/rog.20023, 2013.

Yu, W., Yao, T., Tian, L., Ma, Y., Ichiyanagi, K., Wang, Y., and Sun, W.: Relationships between δ18O in precipitation and air temperature and moisture origin on a south–north transect of the Tibetan Plateau, Atmos. Res., 87(2), 158-169, https://doi.org/:10.1016/j.atmosres.2007.08.004, 2008.

Zhang, C.: Moisture source assessment and the varying characteristics for the Tibetan Plateau precipitation using TRMM, Environ. Res. Lett., 15(10), 104003, https://doi.org/:10.1088/1748-9326/abac78, 2020.

Zhang, C., Tang, Q., and Chen, D.: Recent changes in the moisture source of precipitation over the Tibetan Plateau, J. Climate, 30(5), 1807-1819, https://doi.org/:10.1175/JCLI-D-15-0842.1, 2017.

Zhang, C., Tang, Q. H., Chen, D. L., van der Ent, R. J., Liu, X. C., Li, W. H., and Haile, G. G.: Moisture Source Changes Contributed to Different Precipitation Changes over the Northern and

Southern Tibetan Plateau, J. Hydrometeorol., 20(2), 217-229, https://doi.org/:10.1175/Jhm-D-18-0094.1, 2019a.

Zhang, Y., Huang, W., and Zhong, D.: Major Moisture Pathways and Their Importance to Rainy Season Precipitation over the Sanjiangyuan Region of the Tibetan Plateau, J. Climate, 32(20), 6837-6857, https://doi.org/:10.1175/jcli-d-19-0196.1, 2019b.

Zhao, H. B., Xu, B. Q., Yao, T. D., Wu, G. J., Lin, S. B., Gao, J., and Wang, M.: Deuterium excess record in a southern Tibetan ice core and its potential climatic implications, Climate Dyn., 38(9-10), 1791-1803, https://doi.org/:10.1007/s00382-011-1161-7, 2012.

---

## Author Response (AR1)

**Response to Anonymous Referee #1**

**General comments:** Despite the moisture sources of the Tibetan Plateau (TP) have been basically revealed by several researches on the basis of different methods, the further study of the moisture sources over TP from various dataset is necessary. In this manuscript, the authors quantified the absolute and relative contributions of oceanic moisture sources over TP based on a moisture tracking model and the various atmospheric reanalysis products. The methods in the manuscript is generally effective, while the moisture tracking method in this study still have non-ignorable uncertainties and need proper evaluation. In science, the novel contribution of this study is not clear due to most of the conclusions have been revealed in previous studies. Therefore, I would recommend that the manuscript need major revision before accepted by HESS. Below are my specific comments.

**Response:** Thank you for your valuable comments and suggestions.

For the uncertainties of the moisture tacking method, in the revision, we further explained (1) why we chose the WAM-2layer model and (2) how we dealt with the uncertainties of this model. The clarifications have been strengthened in the revised manuscript. Please see our detailed response to your specific comment #1.

For the novelty of this study, in the revision, we more clearly pointed out the unique contributions of this paper by (1) summarizing knowledge gaps and (2) thoroughly comparing this study with existing moisture tracking studies in the TP (see Table AC1). Please see our detailed response to your specific comment #2.

**Specific comments:**

1. In this study, there are several approximation in the Eulerian moisture tracking method (Van der Ent et al., 2010; Van der Ent, 2014), which induce non-ignorable uncertainties of the moisture sources calculations. For example, it only can resolve two vertical layers in the model and does not consider all the water substances (water vapour, cloud droplets, cloud ice, rain, and snow) and all the physical processes that the moisture undergo in the model, eg. deep convection, shallow convection, cloud macrophysics, cloud microphysics, diffusion etc. It is not the best one. In fact, the detailed quantified moisture models have been developed. In the references in around line 70, I suggest the authors pertinently evaluate previous studies and properly evaluate Van der Ent' (2014) method in section 2.1. Also need indicate the uncertainties of this method in the manuscript.

**Response:** Thank you for the comment and suggestion. We would like to address your concern from the following two aspects.

**(1) Why we chose to use the WAM-2layer model over the study area.**

Firstly, the WAM (Water Accounting Model) has already been widely used in moisture tracking in the TP region, for example, in the central-western TP (*Zhang et al.*, 2017), in the Endorheic TP (*Li et al.*, 2019), between the southern and northern parts of the TP (*Zhang et al.*, 2019a), and in the entire TP (*Guo et al.*, 2019; *Zhang*, 2020). The quantified moisture tracking results with WAM are generally consistent with other commonly used models, such as FLEXPART, HYSPLIT, LAGRANTO, QIBT, and CAM (see Table AC1 for a detailed

comparison). This cross-region, cross-model comparison showcases the reliability and robustness of WAM model for moisture tracking over the study area.

**Table AC1.** Summary of numerical moisture tracking studies in the TP region.

| Reference | Study area | Time period | Model | Data | Main conclusions |
|---|---|---|---|---|---|
| Chen et al. (2012) | TP | 2005–2009 (summer) | FLEXPART | NCEP/GFS | The ocean source could extend from the Arabian Sea to the Southern Hemisphere. |
| Sun and Wang (2014) | Grassland on eastern TP | 2000–2009 | FLEXPART | NCEP-CFSR | During the warm (cold) season, oceanic moisture is mainly from the Arabian Sea and Bay of Bengal (areas surrounding the Arabian Peninsula). |
| Zhang et al. (2017) | Central-western TP | 1979–2013 | WAM | ERA-I, NCEP-2 | More than 21% of the moisture comes from oceans. |
| Huang et al. (2018) | Southeastern TP | 1979–2016 (winter extreme precipitation) | LAGRANTO | ERA-I | About 18% of the moisture comes from oceans. |
| Pan et al. (2018) | Southern/northern TP | 1982–2014 | CAM | MERRA | During summer, the Indian Ocean supplies about 28.5% of the moisture to the southern TP. |
| Chen et al. (2019) | Four areas in TP | 1980–2016 (May–August) | FLEXPART | ERA-I | The northwestern TP and northeastern TP are less affected by the Indian monsoon moisture. |
| Guo et al. (2019) | TP | 1979–2015 | WAM-2layers | ERA-I | Indian Ocean and Pacific Ocean account for 24% and 2% of the moisture contribution, respectively. |
| Li et al. (2019) | Endorheic TP | 1979–2015 | WAM-2layers | ERA-I, MERRA-2, JRA-55 | 24%–30% of the moisture comes from oceans. |
| Qiu et al. (2019) | Three areas in TP | 1979–2016 (winter extreme precipitation) | LAGRANTO | ERA-I | Moisture contributions of the Arabian Sea to the intense precipitation in the western, south-central, and southeastern TP are 9.2%, 6.9%, and 1.1%, respectively. |
| Xu and Gao (2019) | Southeastern TP | 1982–2011 (April–September) | QIBT | ERA-I | Only 2% of the moisture originates from the oceanic source. |
| Zhang et al. (2019a) | Southern/northern TP | 1979–2016 | WAM-2layers | ERA-I | Northwestern (southeastern) source contributes ~39% (~51%) of the moisture in the northern (southern) TP. |
| Zhang et al. (2019b) | Sanjiangyuan Region | 1960–2017 (June–September) | HYSPLIT, HDBSCAN | NNR1 | About 51% (54%) of the medium to heavy precipitation is influenced by the northwestern (southern) source. |
| Liu et al. (2020) | Western TP | 1979–2018 (winter) | HYSPLIT | ERA-I | About 57% of the moisture comes from the Arabian Sea, Arabian Peninsula, and northern Indian Ocean. |
| Ma et al. (2020) | Seven areas in TP | 1961–2015 (summer extreme event) | HYSPLIT | NCEP/NCAR | About 75% of the moisture for extreme precipitation in the southeastern TP comes from the Bay of Bengal. |

| Yang et al. (2020) | Southeastern TP | 1980–2016 (June–September) | FLEXPART | ERA-I | 30% of the moisture comes from oceans. |
|---|---|---|---|---|---|
| Zhang (2020) | TP | 1998–2018 | WAM-2layers | ERA-I, TRMM | The southeastern source from the TP to the western Indian Ocean accounts for 32% of the moisture contribution. |
| LI et al. (2022) | Seven basins in TP | 1979–2015 | WAM-2layers | ERA-I, MERRA-2, JRA-55 | Oceanic moisture accounts for 24%–30% of the moisture in different basins of the TP. |

Secondly, Lagrangian models (e.g., FLEXPART and HYSPLIT) concern the movement of 'air particles' in the atmosphere, thus, the identification of precipitation and evaporation events mainly relies on the dynamic humidity information of each air particle (*Tuinenburg and Staal*, 2020). The detailed methods have been introduced in *Sodemann et al.* (2008) ('moisture source attribution' method) and *Sun and Wang* (2014) ('areal source–receptor attribution' method). In comparison, Eulerian models (i.e., WAM-2layers) focus on the water balance of fixed grids, which enables us to track the precipitation and evaporation moisture separately based on the mass balance principle. This results in different computational costs for long-term studies. In Lagrangian models, researchers generally use a tracking period of about 10 days (the average residence time of moisture in the atmosphere) for a single release of air particles. For long-term experiments as in this work (1979–2020), Lagrangian methods can consume relatively higher computational resources if one continuous release particles from the target region during the period (or releasing a large amount of air particles from all potential source regions at once). Therefore, considering the need for long-term precipitation/evaporation moisture tracking, the WAM-2layer is more suitable in this study.

Thirdly, the model developers of the WAM-2layers have verified the availability of this model at both global and regional scales, by the comparisons with the 'RCM-tag' (MM5, the Fifth-Generation Mesoscale Model) model and the '3D-Trajectories' (QIBT, quasi-isentropic back-trajectory) models (*Van der Ent et al.*, 2013). This comparison has suggested the reliability of WAM-2layers model in moisture tracking.

**(2) How we dealt with the uncertainties of the model.**

Firstly, as mentioned by the reviewer, the model contains only two layers. The two layers are set to adequately deal with the wind shear in the upper air, but this does not affect the accuracy in calculating the $\partial(S_k u))/\partial_x$ and $\partial(S_k u))/\partial_x$ in Equation (1) in the manuscript. In fact, a total of 17 layers of wind fields and specific humidity were used in the model to separate these two model layers. In addition, we also downloaded the total column moisture and vertically integrated moisture fluxes over all tracking areas to revise the calculations of moisture transport in the model.

Secondly, we have considered all possible phases of water in the atmosphere in ERA-Interim and MERRA-2, which contains water vapor, cloud liquid water, and cloud frozen water. One exception is JRA-55, for which we did not consider the cloud liquid/frozen water, as it is not available.

Thirdly, we totally agree with the reviewer that some physical processes, such as the deep convection, shallow convection, cloud macrophysics, cloud microphysics, and diffusions, are not considered in the model. However, the core function of the WAM-2layers is the dynamic reproduction of the moisture transport processes with the input datasets. An analysis at the original resolutions of the input datasets will largely limit uncertainties to input

datasets themselves. We acknowledge that for analyses at a higher spatial-temporal resolution, more physically based models might be more accurate (e.g., WRF-WVT). In this work, all analyses were conducted at the original spatial resolution of the input datasets (1°×1°). To better capture the vertical exchanges due to convection, turbulence, and re-evaporation and minimize the water balance losses between different model layers in a higher temporal resolution in the WAM-2layers, the gross vertical flow is set to 4 times the vertical flow in the net flow direction and 3 times the vertical flow in the opposite direction according to the studies from *van der Ent et al.* (2014) and *Findell et al.* (2019).

Fourthly, to better demonstrate the reliability of our conclusions and potential uncertainties, we used three reanalysis products for moisture tracking over the study area. We have ensured that all relevant conclusions are supported by results using different reanalysis products.

Based on the above descriptions, we have strengthened the relevant advantages and uncertainties of the WAM-2layers model in Section 2.1 of our revised manuscript, and thoroughly evaluated previous studies in moisture tacking over the TP as Table S1 (i.e., Table AC1).

2. In science, the novel contribution of this study is not clear. The absolute and relative contributions of moisture sources, including oceanic source over TP have been quantitatively revealed. I suggest the authors focus on the comparisons of moisture sources evaluation based on the various atmospheric reanalysis products. The relationship between model oceanic source and isotope δ18O is interesting.

**Response:** Thanks for your suggestions. To highlight the novelty of this study, we re-summarized the scientific significances from the following two aspects:

**(1) The pressing need to study the oceanic moisture contribution to the TP.**

Firstly, the TP has been considered as a thermal "air pump" that attracts low-latitude oceanic evaporation to the region, particularly under recently altered land-sea thermal gradient between the TP and global oceans (meteorological records revealed that the atmospheric warming rate over the TP was twice that of the global mean). A quantitative, spatial and temporal evaluation of the oceanic moisture contribution to the TP could help better understand the changing hydrological cycle over the TP and its underlying mechanisms. This part is described in lines 28–46 in our revised manuscript.

Secondly, the interpretations of paleoclimate records in the TP, particularly the $\delta^{18}O$ and $\delta D$ in the precipitation and ice-cores, largely rely on the understanding of different moisture sources for the TP. For example, the $\delta^{18}O$ and $\delta D$ evaporated from oceans are relatively enriched in comparison with the other sources. Different oceanic contributions may link to different isotope values in different climate regions of the TP, which has not been thoroughly explored. This part is mentioned in lines 67–71 in our revised manuscript.

More specifically, we distinguished the moisture contribution of the Indian Ocean (IO) from that of the Western Oceans (WO) in our analyses. These two regions represent the source areas of the Indian summer monsoon and the mid-latitude westerlies (the two core circulation systems dominate the TP's climate), respectively. For example, by using numerous $\delta^{18}O$ measurements from precipitation and ice-core on the TP, Tian et al. (2007), Yao et al. (2013), and numerous isotope-related studies (Tian et al., 2001; Yu et al., 2008;

Hren et al., 2009; Zhao et al., 2012; Joswiak et al., 2013; Ren et al., 2021) empirically identified a line around the 34°–35°N to represent the northward extension of the Indian summer monsoon. In this context, we intend to provide a quantitative view of the region influenced by the Indian monsoon, from the perspective of moisture contributions. This part is included in lines 48–54 and 71–75 in our revised manuscript.

**(2) The novelty of this study as compared with previous moisture tracking studies in the TP.**

In comparison with the traditional synoptic and climatological analyses, the numerical moisture tracking method could quantitatively diagnose the moisture contribution to a target region. In Table AC1 above, we summarize existing studies using numerical moisture tracking in the TP published during the past two decades. Although these studies have quantified the oceanic moisture contribution to different parts of the TP in different seasons after the 1960s, nearly all of them only considered *regional averages* for specific target areas in the TP ('Study area' in Table AC1) with *backward* moisture tracking. The *spatial distribution* of oceanic moisture contribution to the vast TP, e.g., the transition gradient of the moisture transported from the Indian Ocean, is hitherto unclear. To fill this knowledge gap, in this study we leveraged a *forward* moisture tracking method and studied the *spatial distribution* of oceanic moisture contribution over the TP. This part is mentioned in lines 61–65 in our revised manuscript, and the Table AC1 is added as Table S1 in our revised Supplementary.

3. In line 147, I do not think the oceanic sources of the Mediterranean, the Red Sea, and the Persian Gulf, can compared to the Atlantic. They are too small. If say this, please give the quantitative tracking results.

**Response:** Thanks for pointing this out. Figure AC1 below shows the long-term mean contribution of moisture source to the TP precipitation in summer (Figures AC1a–c), in winter (Figures AC1d–f), and on an annual scale (Figures AC1 g–i). Although the spatial extent of the Mediterranean, the Red Sea, and the Persian Gulf is much smaller than that of the Atlantic, as pointed out by the reviewer, their relative contribution to TP precipitation is non-negligible. In fact, the summation of the contributions from these three regions can be greater than the contribution from the Atlantic (Table AC2 below summarizes the relative contributions of these four regions to summer, winter, and annual precipitation over the TP based on ERA-Interim dataset).

The Figures AC1a–f, Figures AC1g–i, and Table AC2 were shown as Figure S3, Figures S6a–c, and Table S3 in our revised Supplementary. Please see lines 179–184 in our revised manuscript for the justification.

[Figure]

**Figure AC1.** Long-term mean moisture source of the TP precipitation in summer (a, b, and c), winter (d, e, and f), and on an annual scale (g, h, and i).

**Table AC2.** Relative moisture contribution to the TP from different oceans.

|  | The Atlantic | The Mediterranean | The Red Sea | The Persian Gulf |
|---|---|---|---|---|
| Summer | 1.88% | 1.05% | 0.35% | 0.82% |
| Winter | 13.76% | 8.43% | 5.39% | 4.42% |
| Annual | 4.49% | 2.75% | 1.36% | 1.57% |

4. Please indicate the sub-figures when describe in around line 239.

**Response:** Thanks for the suggestion and sorry for the confusion. We have cited the sub-figures accordingly in the revised Lines 290–291: "Quantitatively, this geographical barrier of the monsoon system reflected in water isotope ratios closely aligns with the 10%–20% isoline of the relative contribution from IO (Figure 4h)."

**References:**

Chen, B., Xu, X. D., Yang, S., and Zhang, W.: On the origin and destination of atmospheric moisture and air mass over the Tibetan Plateau, Theor. Appl. Climatol., 110(3), 423-435, https://doi.org/:10.1007/s00704-012-0641-y 2012.

Chen, B., Zhang, W., Yang, S., and Xu, X. D.: Identifying and contrasting the sources of the water vapor reaching the subregions of the Tibetan Plateau during the wet season, Climate Dyn., 53(11), 6891-6907, https://doi.org/:10.1007/s00382-019-04963-2, 2019.

Findell, K. L., Keys, P. W., van der Ent, R. J., Lintner, B. R., Berg, A., and Krasting, J. P.: Rising Temperatures Increase Importance of Oceanic Evaporation as a Source for Continental Precipitation, J. Climate, 32(22), 7713-7726, https://doi.org/:10.1175/jcli-d-19-0145.1, 2019.

Guo, L., van der Ent, R. J., Klingaman, N. P., Demory, M.-E., Vidale, P. L., Turner, A. G., Stephan, C. C., and Chevuturi, A.: Moisture Sources for East Asian Precipitation: Mean Seasonal Cycle and Interannual Variability, J. Hydrometeorol., 20(4), 657-672, https://doi.org/:10.1175/jhm-d-18-0188.1, 2019.

Hren, M. T., Bookhagen, B., Blisniuk, P. M., Booth, A. L., and Chamberlain, C. P.: δ18O and δD of streamwaters across the Himalaya and Tibetan Plateau: Implications for moisture sources and paleoelevation reconstructions, Earth Planet. Sci. Lett., 288(1), 20-32, https://doi.org/:10.1016/j.epsl.2009.08.041, 2009.

Huang, W., Qiu, T., Yang, Z., Lin, D., Wright, J. S., Wang, B., and He, X.: On the formation mechanism for wintertime extreme precipitation events over the southeastern Tibetan Plateau, J. Geophys. Res.-Atmos., 123(22), 12,692-612,714, https://doi.org/:10.1029/2018JD028921 2018.

Joswiak, D. R., Yao, T., Wu, G., Tian, L., and Xu, B.: Ice-core evidence of westerly and monsoon moisture contributions in the central Tibetan Plateau, J. Glaciol., 59(213), 56-66, https://doi.org/:10.3189/2013JoG12J035 2013.

Li, Y., Su, F., Chen, D., and Tang, Q.: Atmospheric Water Transport to the Endorheic Tibetan Plateau and Its Effect on the Hydrological Status in the Region, J. Geophys. Res.-Atmos., 124(23), 12864-12881, https://doi.org/:10.1029/2019jd031297, 2019.

LI, Y., SU, F., TANG, Q., GAO, H., YAN, D., PENG, H., and XIAO, S.: Contributions of moisture sources to precipitation in the major drainage basins in the Tibetan Plateau, Sci. China-Earth Sci., 65(1674-7313), 1088, https://doi.org/:https://doi.org/10.1007/s11430-021-9890-6, 2022.

Liu, X., Liu, Y., Wang, X., and Wu, G.: Large-Scale Dynamics and Moisture Sources of the Precipitation Over the Western Tibetan Plateau in Boreal Winter, J. Geophys. Res.-Atmos., 125(9), e2019JD032133, https://doi.org/:10.1029/2019JD032133, 2020.

Ma, Y., Lu, M., Bracken, C., and Chen, H.: Spatially coherent clusters of summer precipitation extremes in the Tibetan Plateau: Where is the moisture from?, Atmos. Res., 237, 104841, https://doi.org/:10.1016/j.atmosres.2020.104841, 2020.

Pan, C., Zhu, B., Gao, J., Kang, H., and Zhu, T.: Quantitative identification of moisture sources over the Tibetan Plateau and the relationship between thermal forcing and moisture transport, Climate Dyn., 52(1-2), 181-196, https://doi.org/:10.1007/s00382-018-4130-6, 2018.

Qiu, T., Huang, W., Wright, J. S., Lin, Y., Lu, P., He, X., Yang, Z., Dong, W., Lu, H., and Wang, B.: Moisture Sources for Wintertime Intense Precipitation Events Over the Three Snowy Subregions of the Tibetan Plateau, J. Geophys. Res.-Atmos., 124(23), 12708-12725, https://doi.org/:10.1029/2019jd031110, 2019.

Ren, W., Tian, L., and Shao, L.: Regional moisture sources and Indian summer monsoon (ISM) moisture transport from simultaneous monitoring of precipitation isotopes on the southeastern and northeastern Tibetan Plateau, J. Hydrol., 601, 126836, https://doi.org/:10.1016/j.jhydrol.2021.126836, 2021.

Sodemann, H., Schwierz, C., and Wernli, H.: Interannual variability of Greenland winter precipitation sources: Lagrangian moisture diagnostic and North Atlantic Oscillation influence, J. Geophys. Res.-Atmos., 113(D3), D03107, https://doi.org/:10.1029/2007JD008503, 2008.

Sun, B., and Wang, H.: Moisture sources of semiarid grassland in China using the Lagrangian particle model FLEXPART, J. Climate, 27(6), 2457-2474, https://doi.org/:10.1175/JCLI-D-13-00517.1 2014.

Tian, L., Masson-Delmotte, V., Stievenard, M., Yao, T., and Jouzel, J.: Tibetan Plateau summer monsoon northward extent revealed by measurements of water stable isotopes, J. Geophys. Res.-Atmos., 106(D22), 28081-28088, https://doi.org/:10.1029/2001jd900186, 2001.

Tian, L., Yao, T., MacClune, K., White, J., Schilla, A., Vaughn, B., Vachon, R., and Ichiyanagi, K.: Stable isotopic variations in west China: a consideration of moisture sources, Journal of Geophysical Research: Atmospheres (1984–2012), 112(D10), D10112, https://doi.org/:10.1029/2006JD007718, 2007.

Tuinenburg, O. A., and Staal, A.: Tracking the global flows of atmospheric moisture and associated uncertainties, Hydrol. Earth Syst. Sci., 24(5), 2419-2435, https://doi.org/:10.5194/hess-24-2419-2020, 2020.

Van der Ent, R. J., Tuinenburg, O. A., Knoche, H. R., Kunstmann, H., and Savenije, H. H. G.: Should we use a simple or complex model for moisture recycling and atmospheric moisture tracking?, Hydrol. Earth Syst. Sci., 17(12), 4869-4884, https://doi.org/:10.5194/hess-17-4869-2013, 2013.

van der Ent, R. J., Wang-Erlandsson, L., Keys, P. W., and Savenije, H. H. G.: Contrasting roles of interception and transpiration in the hydrological cycle – Part 2: Moisture recycling, Earth Syst. Dynam., 5(2), 471-489, https://doi.org/:10.5194/esd-5-471-2014, 2014.

Xu, Y., and Gao, Y.: Quantification of Evaporative Sources of Precipitation and Its Changes in the Southeastern Tibetan Plateau and Middle Yangtze River Basin, Atmosphere, 10(8), 428, https://doi.org/:10.3390/atmos10080428, 2019.

Yang, S., Zhang, W., Chen, B., Xu, X., and Zhao, R.: Remote moisture sources for 6-hour summer precipitation over the Southeastern Tibetan Plateau and its effects on precipitation intensity, Atmos. Res., 236, 104803, https://doi.org/:10.1016/j.atmosres.2019.104803, 2020.

Yao, T., Masson-Delmotte, V., Gao, J., Yu, W., Yang, X., Risi, C., Sturm, C., Werner, M., Zhao, H., and He, Y.: A review of climatic controls on δ18O in precipitation over the Tibetan Plateau: Observations and simulations, Rev. Geophys., 51(4), 525-548, https://doi.org/:10.1002/rog.20023, 2013.

Yu, W., Yao, T., Tian, L., Ma, Y., Ichiyanagi, K., Wang, Y., and Sun, W.: Relationships between δ18O in precipitation and air temperature and moisture origin on a south–north transect of the Tibetan Plateau, Atmos. Res., 87(2), 158-169, https://doi.org/:10.1016/j.atmosres.2007.08.004, 2008.

Zhang, C.: Moisture source assessment and the varying characteristics for the Tibetan Plateau precipitation using TRMM, Environ. Res. Lett., 15(10), 104003, https://doi.org/:10.1088/1748-9326/abac78, 2020.

Zhang, C., Tang, Q., and Chen, D.: Recent changes in the moisture source of precipitation over the Tibetan Plateau, J. Climate, 30(5), 1807-1819, https://doi.org/:10.1175/JCLI-D-15-0842.1, 2017.

Zhang, C., Tang, Q. H., Chen, D. L., van der Ent, R. J., Liu, X. C., Li, W. H., and Haile, G. G.: Moisture Source Changes Contributed to Different Precipitation Changes over the Northern and Southern Tibetan Plateau, J. Hydrometeorol., 20(2), 217-229, https://doi.org/:10.1175/Jhm-D-18-0094.1, 2019a.

Zhang, Y., Huang, W., and Zhong, D.: Major Moisture Pathways and Their Importance to Rainy Season Precipitation over the Sanjiangyuan Region of the Tibetan Plateau, J. Climate, 32(20), 6837-6857, https://doi.org/:10.1175/jcli-d-19-0196.1, 2019b.

Zhao, H. B., Xu, B. Q., Yao, T. D., Wu, G. J., Lin, S. B., Gao, J., and Wang, M.: Deuterium excess record in a southern Tibetan ice core and its potential climatic implications, Climate Dyn., 38(9-10), 1791-1803, https://doi.org/:10.1007/s00382-011-1161-7, 2012.

**Response to Referee #2 (Ruud van der Ent):**

**General comments:** The authors analyzed the moisture sources of the Tibetan Plateau using 3 different reanalysis products, a widely used moisture tracking method WAM2layers and additional stable isotope data.

   English editing of the paper is absolutely necessary as there are many small mistakes, but this can easily be solved using and English editing service.

   Scientifically, the paper is clear, but the whole analysis can also be considered rather straightforward meaning that the novelty is somewhat minor. Obviously not all papers have to be major breakthroughs, but it would be nice if the authors could indicate a bit more specific what we now know that we did not know before from other studies that analyzed the moisture sources of the Tibetan Plateau.

   My major comment regards the analysis in subsection 3.3 and the conclusion that a decrease in oceanic moisture contribution resulted in reduced TP precipitation. I have strong reservations with this conclusion since as far as I can see the cause and effect could very well be the other way around. This needs more detailed investigation and possibly less strong conclusions.

   I attach more specific comments as a supplement.

**Response:** We are very grateful for your thorough review and comments, which help improve our manuscript and provide guidance for our future research.

   **1.** We are sorry for our grammatical issues in the first submission. We have thoroughly checked and corrected grammatical errors in the revised manuscript with the help of a native speaker. The English editing service will be adopted if there are still grammatical errors in this revision.

   **2.** For the novelty of this work, in the revision, we have pointed out the unique contributions of this paper by (1) summarizing knowledge gaps in the field and (2) thoroughly comparing this study with existing moisture tracking studies in the TP (see Table AC1 in this response document). Please see our detailed response to specific comment #5.

   **3.** For the analyses of oceanic moisture contribution and precipitation change over the TP in Section 3.3, we have thoroughly revised this section according to your specific comments #15, #16, #19, and #22. Please see our detailed response to specific comment #15.

   Please see our responses to your specific comments below.

**Specific comments:**

1. In line 1 (*Title*).
**Comments:** In this way the title is 'imperative' which probably is not really intended.
**Response:** Thanks for pointing this out. Considering both reviewers' comments, we have changed the title to "Spatial Distribution of Oceanic Moisture Contribution to the Precipitation over the Tibetan Plateau". This title emphasizes the 'Spatial Distribution' which is the core novelty in comparison with previous studies in the TP (please see our detailed response to specific comment #5).

**2. In line 11 (*Although recent accelerated global hydrological cycle*).**

**Comments:** Incorrect English: probably meant: 'although the global hydrological cycle recently accelerated'. Contentwise: what does recently exactly mean?

**Response:** Thanks for pointing this out. We have carefully revised the language issues in our revision. By "recently" we meant in recent years. In this revision, this sentence has been revised to "Although the accelerated global hydrological cycle, the altered sea–land thermal contrast, and the amplified warming rate over the TP during the past several decades are known to have profound effects on the regional water balance, the spatial distribution of oceanic moisture contribution to the vast TP remains unclear." Please see lines 14–16 in our revised manuscript.

**3. In lines 25–26 (*Van der Ent et al., 2010; Trenberth et al., 2011*).**

**Comments:** Aren't there more recent estimates?

**Response:** Thanks for pointing this out. For the description "evaporation from oceans constitutes more than 80% of the global surface evaporation", the value "80%" is from *Trenberth et al.* (2011) which quantified the global oceanic evaporation and terrestrial evapotranspiration using six different reanalysis products (NCEP-NCAR R1/R2, CFSR, C20r, ERA-40, ERA-Interim, JRA-25, and MERRA). We considered this estimate robust and trustworthy.

For the description "evaporation from oceans contributes to about 60% of terrestrial precipitation", the value "60%" is from *Van der Ent et al.* (2010) (one of the earliest studies). Later on, this estimate was mentioned by *Van der Ent and Savenije* (2013). More recently, *Link et al.* (2020) released a dataset on the fate of land evaporation where the information on the sources of precipitation can be extracted based on WAM-2layers. In addition, *Tuinenburg et al.* (2020) released a high-resolution global atmospheric moisture connection dataset based on a Lagrangian moisture tracking model 'UTrack' (*Tuinenburg and Staal*, 2020). They concluded that about 43%–64% of the global terrestrial precipitation was evaporated from oceans.

In the revised manuscript (lines 28–30), this sentence has been changed to "Evaporation from oceans is one of the most important elements in the global hydrological cycle, which constitutes more than 80% of the global surface evaporation and contributes to about half of the terrestrial precipitation". We have also cited the three recent studies: *Gimeno et al.* (2020), *Link et al.* (2020), and *Tuinenburg et al.* (2020).

**4. In line 28.**

**Comments:** Insert "the".

**Response:** Thanks. We have corrected this in the revision (line 33 in the revised manuscript).

**5. In lines 56–57 (*However, the spatial variation of the oceanic moisture contribution from the Himalayas to the inner TP and their historical changes have not been examined yet*).**

**Comments:** It's unclear to me why the authors specifically highlight the oceanic moisture contribution here. In principle there have been several recent studies about the moisture sources of the TP (including the oceanic ones) that have been overlooked here:

Guo, L., van der Ent, R. J., Klingaman, N. P., Demory, M.-E., Vidale, P. L., Turner, A. G.,

Stephan, C. C., and Chevuturi, A.: Moisture Sources for East Asian Precipitation: Mean Seasonal Cycle and Interannual Variability, 20, 657–672, https://doi.org/10.1175/JHM-D-18-0188.1, 2019.

Zhang, C., Tang, Q., Chen, D., van der Ent, R. J., Liu, X., Li, W., and Haile, G. G.: Moisture Source Changes Contributed to Different Precipitation Changes over the Northern and Southern Tibetan Plateau, J. Hydrometeorol., 20, 217–229, https://doi.org/10.1175/JHM-D-18-0094.1, 2019.

Not only would a citation to these works be appropriate, but also: 1) How does this paper add to what we already know from the aforementioned papers? 2) How do the results from this paper compared to the findings of the aforementioned papers?

**Response:** Thanks for the comments. We would like to address your concern from the following two aspects:

**(1) The pressing need to study the oceanic moisture contribution to the TP:**

Firstly, the TP has been considered a thermal "air pump" that attracts low-latitude oceanic evaporation to the region, particularly considering the altered land-sea thermal gradient between the TP and global oceans in recent years (meteorological records revealed that the atmospheric warming rate over the TP was twice that of the global mean). A quantitative, spatiotemporal evaluation of the oceanic moisture contribution to the TP could help better understand the changing hydrological cycle over the TP and its underlying mechanisms. This part is described in lines 28–46 in our revised manuscript.

Secondly, the interpretations of paleoclimate records in the TP, particularly the $\delta^{18}$O and $\delta$D in the precipitation and ice-cores, largely rely on the understanding of different moisture sources for the TP. For example, the $\delta^{18}$O and $\delta$D evaporated from oceans are relatively enriched in comparison with the other sources, and the precipitation contains relatively low isotope values when strong convection activities occur along the moisture transport processes. Different oceanic contributions may link to different isotope values in different climate regions of the TP, which has not been thoroughly explored. This part is mentioned in lines 67–71 in our revised manuscript.

More specifically, we distinguished the moisture contribution to the TP precipitation from the Indian Ocean (IO) and the Western Oceans (WO). These two regions represent the source areas of the Indian summer monsoon and the mid-latitude westerlies (the two core circulation systems that dominate the TP's climate). For example, using $\delta^{18}$O measurements from precipitation and ice-core on the TP, Tian et al. (2007), Yao et al. (2013), and numerous isotope-related studies (Tian et al., 2001; Yu et al., 2008; Hren et al., 2009; Zhao et al., 2012; Joswiak et al., 2013; Ren et al., 2021) empirically identified a line around the 34°–35°N to represent the northward extension of the Indian summer monsoon. In this context, we intend to provide a quantitative view of the region influenced by the Indian monsoon (from the perspective of moisture contributions). This part is included in lines 48–54 and 71–75 in our revised manuscript.

**(2) The novelty of this study when compared with previous moisture tracking studies in the TP.**

In comparison with the traditional synoptic and climatological analyses, the numerical moisture tracking method could quantitatively diagnose the moisture contribution to a target region. In Table AC1 below, we summarize existing studies using numerical moisture

tracking in the TP published during the past two decades. Although these studies have quantified the oceanic moisture contribution to different parts of the TP in different seasons since the 1960s, nearly all of them only considered *regional averages* for specific target areas in the TP ('Study area' in Table AC1) with *backward* moisture tracking. The *spatial distribution* of oceanic moisture contribution to the vast TP, e.g., the transition gradient of the moisture transported from the Indian Ocean, is hitherto unclear. To fill this knowledge gap, in this study we leveraged the *forward* moisture tracking method and studied the *spatial distribution* of oceanic moisture contribution over the TP. This part is mentioned in lines 61–65 in our revised manuscript, and Table AC1 has been added as Table S1 in our revised Supplementary.

**Table AC1.** Summary of numerical moisture tracking studies in the TP region.

| Reference | Study area | Time period | Model | Data | Main conclusions |
|---|---|---|---|---|---|
| Chen et al. (2012) | TP | 2005–2009 (summer) | FLEXPART | NCEP/GFS | The ocean source could extend from the Arabian Sea to the Southern Hemisphere. |
| Sun and Wang (2014) | Grassland on eastern TP | 2000–2009 | FLEXPART | NCEP-CFSR | During the warm (cold) season, oceanic moisture is mainly from the Arabian Sea and Bay of Bengal (areas surrounding the Arabian Peninsula). |
| Zhang et al. (2017) | Central-western TP | 1979–2013 | WAM | ERA-I, NCEP-2 | More than 21% of the moisture comes from oceans. |
| Huang et al. (2018) | Southeastern TP | 1979–2016 (winter extreme precipitation) | LAGRANTO | ERA-I | About 18% of the moisture comes from oceans. |
| Pan et al. (2018) | Southern/northern TP | 1982–2014 | CAM | MERRA | During summer, the Indian Ocean supplies about 28.5% of the moisture to the southern TP. |
| Chen et al. (2019) | Four areas in TP | 1980–2016 (May–August) | FLEXPART | ERA-I | The northwestern TP and northeastern TP are less affected by the Indian monsoon moisture. |
| Guo et al. (2019) | TP | 1979–2015 | WAM-2layers | ERA-I | The Indian Ocean and the Pacific Ocean account for 24% and 2% of the moisture contribution, respectively. |
| Li et al. (2019) | Endorheic TP | 1979–2015 | WAM-2layers | ERA-I, MERRA-2, JRA-55 | 24%–30% of the moisture comes from oceans. |
| Qiu et al. (2019) | Three areas in TP | 1979–2016 (winter extreme precipitation) | LAGRANTO | ERA-I | Moisture contributions of the Arabian Sea to the intense precipitation in the western, south-central, and southeastern TP are 9.2%, 6.9%, and 1.1%, respectively. |
| Xu and Gao (2019) | Southeastern TP | 1982–2011 (April–September) | QIBT | ERA-I | Only 2% of the moisture originates from the oceanic source. |
| Zhang et al. (2019a) | Southern/northern TP | 1979–2016 | WAM-2layers | ERA-I | Northwestern (southeastern) source contributes ~39% (~51%) of the moisture in the northern (southern) TP. |

| Zhang et al. (2019b) | Sanjiangyuan Region | 1960–2017 (June–September) | HYSPLIT, HDBSCAN | NNR1 | About 51% (54%) of the medium to heavy precipitation is influenced by the northwestern (southern) source. |
| Liu et al. (2020) | Western TP | 1979–2018 (winter) | HYSPLIT | ERA-I | About 57% of the moisture comes from the Arabian Sea, the Arabian Peninsula, and the northern Indian Ocean. |
| Ma et al. (2020) | Seven areas in TP | 1961–2015 (summer extreme event) | HYSPLIT | NCEP/NCAR | About 75% of the moisture for extreme precipitation in the southeastern TP comes from the Bay of Bengal. |
| Yang et al. (2020) | Southeastern TP | 1980–2016 (June–September) | FLEXPART | ERA-I | 30% of the moisture comes from oceans. |
| Zhang (2020) | TP | 1998–2018 | WAM-2layers | ERA-I, TRMM | The southeastern source from the TP to the western Indian Ocean accounts for 32% of the moisture contribution. |
| *Li et al.* (2022) | Seven basins in TP | 1979–2015 | WAM-2layers | ERA-I, MERRA-2, JRA-55 | Oceanic moisture accounts for 24%–30% of the moisture in different basins of the TP. |

6. In line 72.

**Comments:** Insert "the".

**Response:** Thanks. We have corrected this in the revision (line 79 in the revised manuscript).

7. In lines 82–84 (*and in comparison with Lagrangian models (e.g., FLEXible PARTicle (FLEXPART) dispersion model and the Hybrid SingleParticle Lagrangian Integrated Trajectory (HYSPLIT) model), the Eulerian grids enable the model to excel in computation speed and to consider moisture budget from precipitation and evaporation separately(Van der Ent et al., 2013; Van der Ent, 2014))*.

**Comments:** This depends very much on what types of tracking runs are being done. Without going in depth in investigating this I would remove these claims entirely.

**Response:** Thank you for the suggestion. We have removed the inappropriate statement regarding computational cost, and revised this sentence as "In comparison with the commonly used Lagrangian models (e.g., the FLEXible PARTicle (FLEXPART) dispersion model and the Hybrid Single-Particle Lagrangian Integrated Trajectory (HYSPLIT) model) that identify precipitation and evaporation events mainly based on the dynamic humidity information of tracked air particles (Tuinenburg and Staal, 2020), Eulerian grids enable the WAM-2layers to consider moisture budget from precipitation and evaporation separately (Van der Ent et al., 2013; Van der Ent, 2014)." Please see lines 95–99 in our revised manuscript.

8. In line 86 (*Equation 1*).

**Comments:** There was an sign error in Van der Ent, 2014, better to write the equations as in Findell et al. (2019)

Findell, K. L., Keys, P. W., van der Ent, R. J., Lintner, B. R., Berg, A., and Krasting, J. P.: Rising Temperatures Increase Importance of Oceanic Evaporation as a Source for Continental Precipitation, J. Clim., 32, 7713–7726, https://doi.org/10.1175/JCLI-D-19-0145.1, 2019.

**Response:** Thanks for your reminder. The Equation (1) was corrected to $\frac{\partial S_{g,lower}}{\partial t} = -\frac{\partial(S_{g,lower}u)}{\partial x} - \frac{\partial(S_{g,lower}v)}{\partial y} + E_g - P_g \pm F_{v,g}$ for forward moisture tracking in WAM-2layers in the lower layer. In addition, all the text description relevant to this equation has been revised. Please see lines 103–105 in our revised manuscript.

9. In lines 91–92 (*Due the existence of residual $\xi_k$, the closure of the model is defined by a ratio of residuals between the two layers, i.e., $\xi_{top}/S_{top} = \xi_{bottom}/S_{bottom}$*).
**Comments:** That is not the definition of closure, but an assumption that is used in order to calculate the vertical flux (see: van der Ent et al., (2014), appendix B).
   van der Ent, R. J., Wang-Erlandsson, L., Keys, P. W., and Savenije, H. H. G.: Contrasting roles of interception and transpiration in the hydrological cycle - Part 2: Moisture recycling, 5, 471–489, https://doi.org/10.5194/esd-5-471-2014, 2014.
**Response:** Thanks for pointing this out. We have thoroughly inspected all the incorrect or improper descriptions in the method section. Due to the modification of the water balance equation in forward moisture tracking (Equation 1), and according to the relevant descriptions in Van der Ent et al. (2013), van der Ent et al. (2014), and Findell et al. (2019), we revised this part to "The 'well-mixed' assumption is applied in this model, which means that the precipitation is assumed to be immediately removed from the atmosphere in the tracking process (i.e., $P_g/P = S_g/S$, where $P$ and $S$ are total precipitation and total column atmospheric moisture storage, respectively). The two vertical layers in the model are set to deal with the wind shear in the upper air. To better capture the vertical exchanges due to convection, turbulence, and re-evaporation and to minimize the water balance losses between the two layers, the gross vertical flow is set to 4 times the vertical flow in the net flow direction and 3 times the vertical flow in the opposite direction. Although this is a simplification of the turbulent moisture exchange, physically reasonable results have been obtained in previous studies, and the general tracking has been validated against online 3D tracking models (Van der Ent et al., 2013; van der Ent et al., 2014; Findell et al., 2019)." Please see lines 106–113 in our revised manuscript.

10. In line 94 (*1°×1°, and the time step is set as 0.25 h*).
**Comments:** This is at a higher resolution than Van der Ent et al. (2014), who used a 1.5 arcdegree resolution. Yet, the authors have chosen the same time step. This may lead to instable and spurious results at high latitudes or at least internal model corrections to maintain stability.
**Response:** Thanks for this comment. We have tested the sensitivity of our results to the selection of different time steps. Below is an example, which compares the results of two simulations using a 15-min time step (0.25 h) and a 10-min time step. As suggested in Figure AC1a and b, visually the results of the mean annual oceanic moisture contribution to the TP with different time steps are nearly identical. This is also confirmed by the differences between these two runs, as shown in Figure AC1c and d. Discrepancies in moisture tracking results induced by different time steps mainly appear in the western TP (Figure AC1c) but are very minor (~1 mm on the annual scale). The relative differences are below 1% in the TP on

the annual scale (Figure AC1d). This suggests the stability of using different time steps in the study area. Please see lines 116–118 in our revised manuscript.

[Figure]

**Figure AC1.** Simulations of mean annual oceanic moisture contribution to the TP with (a) 0.25-h (15-min) time step and (b) 10-min time step, and (c) the absolute difference (mm year$^{-1}$) and (d) the relative difference (%) between these two simulations.

11. In line 95 (*as around 812 hPa*).
**Comments:** But varying with surface pressure?! This is a very important detail.
**Response:** We have revised this sentence as "the vertical separation between the two layers is prescribed as ~812 hPa at the normal atmospheric pressure (Van der Ent et al., 2013). Note that the atmospheric pressure of the vertical separation varies with different surface pressure (the "half-level" pressure in different reanalysis products is defined as $P_{k-1/2} = A_{k-1/2} + B_{k-1/2}P_s$, where $P_s$ is surface pressure, k represents different model levels, and the values of $A_{k-1/2}$ and $B_{k-1/2}$ are defined independently for different reanalysis datasets)." Please see lines 119–122 in our revised manuscript.

12. In lines 102–103 (*The ocean and land distributions were defined according to the 1°×1° gridded land-sea mask from ERA-Interim*).
**Comments:** The land-sea mask of ERA-Interim considers lakes to be 'sea', but it does not makes sense to consider them 'ocean' in this tracking study in my opinion.
**Response:** Thanks for pointing this out. In our simulation, we removed all inland large lakes (considered as 'sea' in the ERA-Interim), for example, the Caspian Sea and the Black Sea. The final land-sea mask with the 1°×1° spatial resolution used in this work is shown in Figure AC2. In the revision, we have revised the description about the land-sea mask in lines 129–132, and added Figure AC2 to the Supplementary as Figure S2.

[Figure]

**Figure AC2.** The land-sea mask used in this study with 1°×1° spatial resolution (the blue area represents oceans).

13. In line 109 (*6h, 17 layers*).
**Comments:** What kind of layers?
**Response:** Table AC2 summarizes the selected 17 model layers in three reanalysis products used in this study. This table has been added to the Supplementary as Table S2 in our revised manuscript (lines 139–141).

**Table AC2.** Summary of the selected 17 model layers in three reanalysis products.

| | Model layers (from the surface to the upper atmosphere) | | |
| --- | --- | --- | --- |
| | ERA-Interim | MERRA-2 | JRA-55 |
| 1 | 60 | 72 | 1 |
| 2 | 59 | 71 | 2 |
| 3 | 58 | 70 | 3 |
| 4 | 57 | 69 | 4 |
| 5 | 56 | 68 | 5 |
| 6 | 55 | 67 | 6 |
| 7 | 54 | 66 | 7 |
| 8 | 51 | 65 | 9 |
| 9 | 48 | 61 | 12 |
| 10 | 47 | 59 | 14 |
| 11 | 44 | 55 | 17 |
| 12 | 41 | 52 | 20 |
| 13 | 38 | 49 | 23 |
| 14 | 35 | 46 | 26 |
| 15 | 32 | 44 | 29 |
| 16 | 27 | 40 | 34 |
| 17 | 17 | 28 | 44 |

14. In line 123 (*in mm and %*).
**Comments:** mm per what? precipitation is a flux with dimension of length x Time-1, where x is 1,2 or 3; percent of what? of the total local evaporation or of the total sink precipitation or something else?
**Response:** Sorry for the confusion. The unit "mm" represents mm per year, per season, or per month in different parts of the study. The unit % represents the percentage of the total sink

precipitation. These have been declared in our revision (lines 154–155), and all the ambiguous units have been corrected in figures or their captions in our revised manuscript. Nevertheless, the subplots of several figures (e.g., Figures 1, 3, and 4) use both mm year$^{-1}$ and mm season$^{-1}$ while sharing one colorbar. To avoid confusion, we have included their specific units in the captions of these figures but kept the unit mm in colorbars.

**Comments:** This seems to be one of the core results yet cause and effect could be entirely reversed, meaning precipitation on TP declines and as a consequence the oceanic contribution also drops, possibly keeping exactly the same ratio. However, it may also be that the evaporation of the ocean has dropped or that the source area has changed. Simply looking at similarities in Figure 5 is insufficient proof in my opinion. Moreover, this subsection discusses many results in the supplement, but if they are discussed at length they should be in the main text instead.

**Response:** Thanks for this comment. In this revision, we performed additional analyses and concluded that the decreased oceanic contribution is mainly induced by precipitation decrease over TP. We have thoroughly revised Section 3.3 to include new analyses and to improve the clarity, following the structure below:

   **a.** Analyze the long-term trends of oceanic moisture contribution to the TP (Figure AC3), and point out that decreased oceanic moisture contribution was found mainly around the southeastern TP (i.e., the Brahmaputra Canyon region, which has long been considered the most important moisture transport channel for the TP (*Hren et al.*, 2009)). More specifically, the moisture contributions of both the monsoon-dominated Indian Ocean (IO) and the westerlies-dominated Western Oceans (WO) decreased over time around the southeastern TP (Figure AC4).

[Figure]

**Figure AC3.** Trends of oceanic moisture contribution to the TP region with (a) ERA-Interim (1979–2015), (b) MERRA-2 (1980–2015), and (c) JRA-55 (1979–2015).

[Figure]

**Figure AC4.** Trends of oceanic moisture contribution to the TP region from the Indian Ocean (IO, a–c) and the western oceans (WO, d–f) with ERA-Interim (1979–2015), MERRA-2 (1980–2015), and JRA-55 (1979–2015).

**b.** Examine the trends of oceanic moisture contribution by detecting the changes in oceanic evaporation, precipitation, the horizontal wind fields, and the updraft around the TP region. According to the reviewer's suggestion, we calculated the long-term trends of global evaporation and precipitation during 1979–2015 (Figure AC5). Most of the oceanic sources exhibited *enhanced* evaporation, so the reduced oceanic evaporation is unlikely the cause. However, the enhanced oceanic moisture may still lose significantly before reaching the TP due to increased precipitation along the transport pathway, particularly when the moisture travels across the Indian Subcontinent and the Bay of Bengal (Figure AC5b). In addition, we analyzed the inter-annual trends of zonal (u) and meridional (v) wind at 700 hPa and 300 hPa and vertical velocity at 300 hPa (Figure AC6). Significantly weakened eastward and northward winds in the lower atmosphere (700 hPa) are found around the southeastern TP (the weakening of horizontal wind fields in the 300 hPa is not statistically significant in the region). This indicates decreased moisture transport to the region in the lower atmosphere. At the same time, significantly decreased upward motion in the higher atmosphere (300 hPa, Figure AC6e) was found in the southeastern TP. This further verifies the decreased condensation of moisture to form precipitation in this region.

[Figure]

**Figure AC5.** Long-term trends of annual evaporation (a) and precipitation (b) over possible source regions during 1979–2015. Stippling indicates regions with statistically significant trends ($p < 0.05$)

[Figure]

**Figure AC6.** Trends of (a) u-wind at 700 hPa (positive denotes enhanced eastward wind), (b) v-wind at 700 hPa (positive denotes enhanced northward wind), (c) u-wind at 300 hPa (positive denotes enhanced eastward wind), (d) v-wind at 300 hPa (positive denotes enhanced northward wind), and (e) vertical velocity at 300 hPa (positive denotes decreased upward motion) around the TP region during 1979–2015. Stippling indicates regions with statistically significant trends ($p < 0.05$).

**c.** The spatial patterns of the long-term trends in precipitation and oceanic moisture contribution over the TP are similar (Figures AC3 and AC5b). To further investigate whether the change in oceanic moisture contribution is due to precipitation change, we carried out additional backward moisture tracking over the southeastern TP (Figure AC7a). The southeastern TP (SETP) was defined as the purple rectangle in Figure AC3, where the oceanic moisture contribution and the precipitation both show decreasing trends during 1979–2015. The spatial distribution of the trends in moisture contributions to the SETP during the same period is shown in Figure AC7b.

As shown in Figure AC7b, the moisture contributions of both the westerlies-dominated western sources and the monsoon-dominated southern sources to the SETP decreased over time, and most source regions that experienced substantial decreases are over land. Meanwhile, only few areas in the southwestern slope of the Himalayas and the southwestern corner of the TP show increased moisture contribution to the SETP (Figure AC7b). Therefore, its inappropriate to conclude that the decreased oceanic moisture contribution dominate the precipitation change over the SETP, although they happen to have similar spatial patterns.

Please see lines 225–259 for the revised Section 3.3 in our revised manuscript. We have added additional figures relevant to our analysis in this section.

[Figure]

**Figure AC7.** (a) Long-term mean moisture source to precipitation in the SETP and (b) the relevant trends of moisture contributions during 1979–2015 simulated using WAM-2layers driven by ERA-Interim. The blue rectangles represent the SETP. Stippling indicates regions with statistically significant trends ($p < 0.05$).

16. In line 212–213 (*Here we further reveal that this dipole pattern is driven by the changes in oceanic moisture contribution (particularly from IO)*).

**Comments:** As said before cause and effect may not be so easy to separate. Moreover, the authors should give an explanation of why the oceanic moisture contribution drops: which can be: less TP precipitation overall, less oceanic evaporation, changing pattern, ...

**Response:** Thanks for the comment. Based on our additional analyses above, we concluded that the spatial pattern of oceanic moisture contribution change is mainly due to precipitation change over the TP. These new analyses have been included in the thoroughly revised Section 3.3. Please see our response to comment #15 above for details.

17. In line 222 (*The strongest relationship is found between precipitation δ18O and relative oceanic moisture contribution from IO (Figure 6)*).

**Comments:** These are interesting plots, but these relationships should be quantified with at least a correlation metric.

**Response:** Thanks. Per your comments, we calculated the correlation coefficients between precipitation $\delta^{18}O$ and the relative oceanic moisture contribution of IO for the 19 stations on the TP (Table AC3). Note that the length of available isotope data for some stations is < 10 months. To ensure the robustness of correlations, we only calculated correlation coefficients for stations with available isotope data longer than 10 months. In Table AC3, nearly all precipitation $\delta^{18}O$ series are negatively correlated with the relative oceanic moisture contributions from IO, particularly for the westerlies-domain stations where all correlations are statistically significant ($p < 0.05$). These conclusions are consistent with our description in Section 3.4. Please see lines 271 in our revised manuscript (The Table AC3 has been added as Table S4 in the revised Supplementary).

**Table AC3.** Correlation coefficients between monthly precipitation $\delta^{18}O$ and the relative oceanic moisture contribution from IO for 19 stations (derived from ERA-Interim, MERRA-2, and JRA-55, respectively). '*' represents statistically significant correlation coefficients ($p < 0.05$).

| | Model layers (from the surface to the upper atmosphere) | | |
|---|---|---|---|
| | ERA-I | MERRA-2 | JRA-55 |

|  |  |  |  |  |
|---|---|---|---|---|
| Monsoon domain | 1.Nyalam | −0.65* | −0.18 | −0.51 |
|  | 2.Zhangmu | - | - | - |
|  | 3.Dingri | - | - | - |
|  | 4.Larzi | - | - | - |
|  | 5.Baidi | −0.37 | −0.42 | −0.41 |
|  | 6.Wengguo | - | - | - |
|  | 7.Dui | −0.38 | −0.49 | −0.33 |
|  | 8.Lhasa | −0.62* | −0.44 | −0.52 |
|  | 9.Yangcun | - | - | - |
|  | 10.Nagqu | −0.39 | −0.18 | −0.05 |
|  | 11.Lulang | −0.44 | −0.12 | −0.29 |
|  | 12.Nuxia | - | - | - |
|  | 13.Bomi | −0.05 | 0.30 | 0.16 |
| Transition domain | 14.Shiquanhe | - | - | - |
|  | 15.Gaize | −0.73* | −0.52* | −0.36 |
|  | 16.Tuotuohe | −0.80* | −0.63* | −0.36 |
|  | 17.Yushu | −0.06* | 0.04 | 0.32 |
| Westerlies domain | 18.Taxkorgen | −0.87* | −0.87* | −0.84* |
|  | 19.Delingha | −0.84* | −0.84* | −0.75* |

18. In line 228–231 (*Note the mismatches between summer peaks of relative moisture from IO and low $\delta^{18}O$ values in autumn at Lulang, Nuxia, and Bomi near the Brahmaputra Canyon, which is likely attributable to the impact of moisture transported from southeast Asia or the Pacific Ocean driven by the trough embedded in the southern branch of the westerlies(Cai and Tian, 2020)).*

**Comments:** These could be investigated in more detail with moisture tracking rather than simply relying on the Cai and Tian (2020) study.

**Response:** Thanks. Per your suggestion, we conducted additional moisture tracking for monthly moisture sources of the SETP (blue rectangle in Figure AC8, which contains Lulang, Nuxia, and Bomi stations) near the Brahmaputra Canyon in Figure AC8. From June to September, moisture sources gradually extended to southeast Asia and the western Pacific Ocean. This is in line with the finding of *Cai and Tian* (2020). Please see lines 280–282 in our revised manuscript (Figure AC8 has been added as Figure S18 in the revised Supplementary).

[Figure]

**Figure AC8.** Mean monthly moisture sources of precipitation in the SETP simulated using WAM-2layers driven by ERA-Interim (1979–2015). The blue rectangle represents the SETP.

19. In line 254 (*is more consistent with precipitation patterns*).

**Comments:** than what?

**Response:** Sorry for the confusion. We have revised this sentence as "… the absolute contribution of oceanic moisture, when compared with relative contribution, is more consistent with the precipitation patterns …". Please see Lines 304–305 in the revised manuscript.

20. In line 266 (*Data availability*).

**Comments:** In my understanding the authors should here nowadays provide links to where their data (to reproduce their figures) can be found.

The availability of the forcing data can be described in methods, acknowledgements and/or references.

Code availability of the original and adapted WAM2layers model is entirely missing.

**Response:** Thanks for the comments. Considering the size of the data, we will make the data that support the findings of this study available upon reasonable request. We have declared this in the Data Availability section. In addition, we have detailed the datasets and the code of WAM-2layers used in this work in the Data Availability section (lines 317–328 in the revised manuscript), and included them in the reference list:

The ERA-Interim dataset can be downloaded from the official website of the European

Centre for Medium-Range Weather Forecasts (ECMWF): https://www.ecmwf.int/en/forecasts/datasets/reanalysis-datasets/era-interim (ECMWF, 2017). The MERRA-2 dataset is available at https://gmao.gsfc.nasa.gov/reanalysis/MERRA-2/ (NASA Goddard Earth Sciences Data and Information Services Center, 2018), which is managed by the Goddard Earth Sciences Data and Information Services Center (GES DISC), National Aeronautics and Space Administration (NASA). The JRA-55 product was developed by the Japan Meteorological Agency and can be downloaded from https://jra.kishou.go.jp/ (Japan Meteorological Agency, 2018). The ERA5 dataset can be downloaded from the Copernicus Climate Change Service (C3S) Climate Date Store (CDS): https://cds.climate.copernicus.eu/ (Copernicus Climate Change Service CDS, 2021). The TNIP δ18O data can be downloaded from the National Tibetan Plateau/Third Pole Environment Data Center: http://data.tpdc.ac.cn (National Tibetan Plateau Data Center, 2021). The code of WAM-2layers (v2.4.08) is available at https://github.com/ruudvdent/WAM2layersPython (van der Ent, 2022). The data generated in this study are available from the corresponding authors upon reasonable request.

References added in this revision:

European Centre for Medium-Range Weather Forecast (ECMWF): The ERA-Interim reanalysis dataset, available at: https://apps.ecmwf.int/datasets/data/interim-full-daily/, last access: 16 May 2017.

NASA Goddard Earth Sciences Data and Information Services Center (GES DISC): Modern-Era Retrospective analysis for Research and Applications, Version 2, available at: https://disc.gsfc.nasa.gov/datasets?project=MERRA-2, last access: 19 June 2018.

Japan Meteorological Agency: JRA-55: Japanese 55-year Reanalysis, Daily 3-Hourly and 6-Hourly Data, Archived at the National Center for Atmospheric Research, Computational and Information Systems Laboratory, available at: https://rda.ucar.edu/datasets/ds628.0/, last access: 19 July 2018.

Copernicus Climate Change Service (C3S) Climate Date Store (CDS): The ERA5 monthly averages data on single levels from 1979 to present, avaible at: https://cds.climate.copernicus.eu/, last access: 11 November 2021.

National Tibetan Plateau Data Center: Data set of $\delta^{18}$O stable Isotopes in Precipitation from Tibetan Network for Isotopes (1991–2008), available at: http://data.tpdc.ac.cn/en/, last access: 3 September 2021.

van der Ent, R. J.: WAM-2layers v2.4.08, available at: https://github.com/ruudvdent/WAM2layersPython, last access: 5 August 2022.

**21. In line 426 (*Figure 2*).**
**Comments:** mm/month. if the x-axis represents month than write the abbreviations of the months instead of 1 – 12
**Response:** Thanks. We have corrected this in our revised manuscript (line 511).

**22. In line 447 (*Figure 5*).**
**Comments:** It would be (more) relevant to look at trends in oceanic moisture contribution with respect to precipitation within the same reanalysis. I now think you're essentially comparing the different precipitation datasets, meaning these plots would have looked the

same for total precipitation without any moisture tracking.

**Response:** Thanks. Per your comments #15, #16, #19, and #22, we have thoroughly revised Section 3.3, which includes the trends in oceanic moisture contribution and precipitation over the TP. Please see our detailed response to your comment #15 above.

**References:**

Cai, Z., and Tian, L.: What Causes the Postmonsoon 18O Depletion Over Bay of Bengal Head and Beyond?, Geophys. Res. Lett., 47(4), e2020GL086985, https://doi.org/10.1029/2020GL086985, 2020.

Chen, B., Xu, X. D., Yang, S., and Zhang, W.: On the origin and destination of atmospheric moisture and air mass over the Tibetan Plateau, Theor. Appl. Climatol., 110(3), 423–435, https://doi.org/10.1007/s00704-012-0641-y 2012.

Chen, B., Zhang, W., Yang, S., and Xu, X. D.: Identifying and contrasting the sources of the water vapor reaching the subregions of the Tibetan Plateau during the wet season, Climate Dyn., 53(11), 6891–6907, https://doi.org/10.1007/s00382-019-04963-2, 2019.

Findell, K. L., Keys, P. W., van der Ent, R. J., Lintner, B. R., Berg, A., and Krasting, J. P.: Rising Temperatures Increase Importance of Oceanic Evaporation as a Source for Continental Precipitation, J. Climate, 32(22), 7713-7726, https://doi.org/10.1175/jcli-d-19-0145.1, 2019.

Gimeno, L., Nieto, R., and Sori, R.: The growing importance of oceanic moisture sources for continental precipitation, Npj Climate and Atmospheric Science, 3(1), 27, https://doi.org/10.1038/s41612-020-00133-y, 2020.

Guo, L., van der Ent, R. J., Klingaman, N. P., Demory, M.-E., Vidale, P. L., Turner, A. G., Stephan, C. C., and Chevuturi, A.: Moisture Sources for East Asian Precipitation: Mean Seasonal Cycle and Interannual Variability, J. Hydrometeorol., 20(4), 657-672, https://doi.org/10.1175/jhm-d-18-0188.1, 2019.

Hren, M. T., Bookhagen, B., Blisniuk, P. M., Booth, A. L., and Chamberlain, C. P.: δ18O and δD of streamwaters across the Himalaya and Tibetan Plateau: Implications for moisture sources and paleoelevation reconstructions, Earth Planet. Sci. Lett., 288(1), 20-32, https://doi.org/10.1016/j.epsl.2009.08.041, 2009.

Huang, W., Qiu, T., Yang, Z., Lin, D., Wright, J. S., Wang, B., and He, X.: On the formation mechanism for wintertime extreme precipitation events over the southeastern Tibetan Plateau, J. Geophys. Res.-Atmos., 123(22), 12,692-612,714, https://doi.org/10.1029/2018JD028921 2018.

Joswiak, D. R., Yao, T., Wu, G., Tian, L., and Xu, B.: Ice-core evidence of westerly and monsoon moisture contributions in the central Tibetan Plateau, J. Glaciol., 59(213), 56-66, https://doi.org/10.3189/2013JoG12J035 2013.

Li, Y., Su, F., Chen, D., and Tang, Q.: Atmospheric Water Transport to the Endorheic Tibetan Plateau and Its Effect on the Hydrological Status in the Region, J. Geophys. Res.-Atmos., 124(23), 12864-12881, https://doi.org/10.1029/2019jd031297, 2019.

Li, Y., Su, F., Tang, Q., Gao, H., Yan, D., Peng, H., and Xiao, S.: Contributions of moisture sources to precipitation in the major drainage basins in the Tibetan Plateau, Sci. China-Earth Sci., 65(1674-7313), 1088, https://doi.org/https://doi.org/10.1007/s11430-021-9890-6, 2022.

Link, A., van der Ent, R., Berger, M., Eisner, S., and Finkbeiner, M.: The fate of land evaporation - a global dataset, Earth System Science Data, 12(3), 1897-1912, https://doi.org/10.5194/essd-12-1897-2020, 2020.

Liu, X., Liu, Y., Wang, X., and Wu, G.: Large-Scale Dynamics and Moisture Sources of the Precipitation Over the Western Tibetan Plateau in Boreal Winter, J. Geophys. Res.-Atmos., 125(9), e2019JD032133,

https://doi.org/10.1029/2019JD032133, 2020.

Ma, Y., Lu, M., Bracken, C., and Chen, H.: Spatially coherent clusters of summer precipitation extremes in the Tibetan Plateau: Where is the moisture from?, Atmos. Res., 237, 104841, https://doi.org/10.1016/j.atmosres.2020.104841, 2020.

Pan, C., Zhu, B., Gao, J., Kang, H., and Zhu, T.: Quantitative identification of moisture sources over the Tibetan Plateau and the relationship between thermal forcing and moisture transport, Climate Dyn., 52(1-2), 181-196, https://doi.org/10.1007/s00382-018-4130-6, 2018.

Qiu, T., Huang, W., Wright, J. S., Lin, Y., Lu, P., He, X., Yang, Z., Dong, W., Lu, H., and Wang, B.: Moisture Sources for Wintertime Intense Precipitation Events Over the Three Snowy Subregions of the Tibetan Plateau, J. Geophys. Res.-Atmos., 124(23), 12708-12725, https://doi.org/10.1029/2019jd031110, 2019.

Ren, W., Tian, L., and Shao, L.: Regional moisture sources and Indian summer monsoon (ISM) moisture transport from simultaneous monitoring of precipitation isotopes on the southeastern and northeastern Tibetan Plateau, J. Hydrol., 601, 126836, https://doi.org/10.1016/j.jhydrol.2021.126836, 2021.

Sun, B., and Wang, H.: Moisture sources of semiarid grassland in China using the Lagrangian particle model FLEXPART, J. Climate, 27(6), 2457-2474, https://doi.org/10.1175/JCLI-D-13-00517.1 2014.

Tian, L., Masson-Delmotte, V., Stievenard, M., Yao, T., and Jouzel, J.: Tibetan Plateau summer monsoon northward extent revealed by measurements of water stable isotopes, J. Geophys. Res.-Atmos., 106(D22), 28081-28088, https://doi.org/10.1029/2001jd900186, 2001.

Tian, L., Yao, T., MacClune, K., White, J., Schilla, A., Vaughn, B., Vachon, R., and Ichiyanagi, K.: Stable isotopic variations in west China: a consideration of moisture sources, Journal of Geophysical Research: Atmospheres (1984–2012), 112(D10), D10112, https://doi.org/10.1029/2006JD007718, 2007.

Trenberth, K. E., Fasullo, J. T., and Mackaro, J.: Atmospheric Moisture Transports from Ocean to Land and Global Energy Flows in Reanalyses, J. Climate, 24(18), 4907-4924, https://doi.org/10.1175/2011JCLI4171.1, 2011.

Tuinenburg, O. A., and Staal, A.: Tracking the global flows of atmospheric moisture and associated uncertainties, Hydrol. Earth Syst. Sci., 24(5), 2419-2435, https://doi.org/10.5194/hess-24-2419-2020, 2020.

Tuinenburg, O. A., Theeuwen, J. J. E., and Staal, A.: High-resolution global atmospheric moisture connections from evaporation to precipitation, Earth Syst. Sci. Data, 12(4), 3177-3188, https://doi.org/10.5194/essd-12-3177-2020, 2020.

Van der Ent, R. J., and Savenije, H. H.: Oceanic sources of continental precipitation and the correlation with sea surface temperature, Water Resour. Res., 49(7), 3993-4004, https://doi.org/10.1002/wrcr.20296, 2013.

Van der Ent, R. J., Savenije, H. H., Schaefli, B., and Steele-Dunne, S. C.: Origin and fate of atmospheric moisture over continents, Water Resour. Res., 46(9), W09525, https://doi.org/10.1029/2010WR009127, 2010.

Van der Ent, R. J., Tuinenburg, O. A., Knoche, H. R., Kunstmann, H., and Savenije, H. H. G.: Should we use a simple or complex model for moisture recycling and atmospheric moisture tracking?, Hydrol. Earth Syst. Sci., 17(12), 4869-4884, https://doi.org/10.5194/hess-17-4869-2013, 2013.

van der Ent, R. J., Wang-Erlandsson, L., Keys, P. W., and Savenije, H. H. G.: Contrasting roles of interception and transpiration in the hydrological cycle – Part 2: Moisture recycling, Earth Syst. Dynam., 5(2), 471-489, https://doi.org/10.5194/esd-5-471-2014, 2014.

Xu, Y., and Gao, Y.: Quantification of Evaporative Sources of Precipitation and Its Changes in the

Southeastern Tibetan Plateau and Middle Yangtze River Basin, Atmosphere, 10(8), 428, https://doi.org/10.3390/atmos10080428, 2019.

Yang, S., Zhang, W., Chen, B., Xu, X., and Zhao, R.: Remote moisture sources for 6-hour summer precipitation over the Southeastern Tibetan Plateau and its effects on precipitation intensity, Atmos. Res., 236, 104803, https://doi.org/10.1016/j.atmosres.2019.104803, 2020.

Yao, T., Masson-Delmotte, V., Gao, J., Yu, W., Yang, X., Risi, C., Sturm, C., Werner, M., Zhao, H., and He, Y.: A review of climatic controls on δ18O in precipitation over the Tibetan Plateau: Observations and simulations, Rev. Geophys., 51(4), 525-548, https://doi.org/10.1002/rog.20023, 2013.

Yu, W., Yao, T., Tian, L., Ma, Y., Ichiyanagi, K., Wang, Y., and Sun, W.: Relationships between δ18O in precipitation and air temperature and moisture origin on a south–north transect of the Tibetan Plateau, Atmos. Res., 87(2), 158-169, https://doi.org/10.1016/j.atmosres.2007.08.004, 2008.

Zhang, C.: Moisture source assessment and the varying characteristics for the Tibetan Plateau precipitation using TRMM, Environ. Res. Lett., 15(10), 104003, https://doi.org/10.1088/1748-9326/abac78, 2020.

Zhang, C., Tang, Q., and Chen, D.: Recent changes in the moisture source of precipitation over the Tibetan Plateau, J. Climate, 30(5), 1807-1819, https://doi.org/10.1175/JCLI-D-15-0842.1, 2017.

Zhang, C., Tang, Q. H., Chen, D. L., van der Ent, R. J., Liu, X. C., Li, W. H., and Haile, G. G.: Moisture Source Changes Contributed to Different Precipitation Changes over the Northern and Southern Tibetan Plateau, J. Hydrometeorol., 20(2), 217-229, https://doi.org/10.1175/Jhm-D-18-0094.1, 2019a.

Zhang, Y., Huang, W., and Zhong, D.: Major Moisture Pathways and Their Importance to Rainy Season Precipitation over the Sanjiangyuan Region of the Tibetan Plateau, J. Climate, 32(20), 6837-6857, https://doi.org/10.1175/jcli-d-19-0196.1, 2019b.

Zhao, H. B., Xu, B. Q., Yao, T. D., Wu, G. J., Lin, S. B., Gao, J., and Wang, M.: Deuterium excess record in a southern Tibetan ice core and its potential climatic implications, Climate Dyn., 38(9-10), 1791-1803, https://doi.org/10.1007/s00382-011-1161-7, 2012.

---

## Author Response (AR2)

**Response to Executive Editor (Prof. Thom Bogaard)**

**Comments to the author:** Dear authors, I received two mixed referee reports. One was generally happy, and has minor points to address. The other reviewer points out that you did not address the review completely. I do not fully agree, as I think you showed the niche of your work sufficiently. Also, I do not think you have to repeat the discussion on Eulerian/Lagrangian approaches in moisture tracking. However, I do think it is appropriate in a scientific debate to discuss that differences in approach exists and what are the strength and weaknesses. This could be addressed a bit more specific.

There is one point I am less happy with. You state: "The data generated in this study are available from the corresponding authors upon reasonable request." You indicate all other researchers sharing data and scripts, whereas you seems reluctant. I do not agree with this statement. I think you should be more open in sharing the data underlying your figures with the community. There are ample open repositories available. So please do so.

**Response:** Thank you very much for handling and reviewing our paper. Per your suggestions, we further strengthened the descriptions about the strength and weaknesses of different approaches in lines 95–102 in our revised manuscript: "In comparison with the commonly used Lagrangian models (e.g., the FLEXible PARTicle (FLEXPART) dispersion model and the Hybrid Single-Particle Lagrangian Integrated Trajectory (HYSPLIT) model) that concern the movement of "air particles" in the atmosphere and identify precipitation and evaporation events mainly based on the dynamic humidity information of the tracked particles, Eulerian models principally focus on moisture transport among fixed grids. In general, Lagrangian models are more accurate and run faster than Eulerian ones for short-term moisture tracking of single grid cells, while Eulerian models are more efficient for long-term moisture tracking over large target regions (Tuinenburg and Staal, 2020). More importantly, the selection of WAM-2layers enables us to consider moisture budget from precipitation and evaporation separately on Eulerian grids (Van der Ent et al., 2013; Van der Ent, 2014)."

We fully agree with your comment on data sharing. We have uploaded the key results of this work as a dataset to the online data repository of the National Tibetan Plateau Data Center (TPDC). The TPDC is the only data center in China with the most complete scientific data for the Tibetan Plateau and surrounding regions. The dataset titled "Dataset of oceanic moisture contribution to precipitation over the Tibetan Plateau simulated by WAM-2 during 1979-2015" is already online and is publicly available at https://data.tpdc.ac.cn/en/data/c6f758cf-6c99-4023-8026-f59e6d3657cb/ (DOI: 10.11888/Atmos.tpdc.272946; see screenshot below). Note that the related literature information will be updated once the paper is published.

**TPDC** National Tibetan Plateau / Third Pole Environment **Data Center**

Keyword(s)           Search

Home  Data  Analysis  Models  News  Submit Data  About

**Dataset of oceanic moisture contribution to precipitation over the Tibetan Plateau simulated by WAM-2 during 1979-2015**

Dataset of oceanic moisture contribution to precipitation over the Tibetan Plateau simulated by WAM-2 during 1979-2015

[Figure]

This is a monthly grided dataset (1°×1°) of the oceanic moisture contribution to precipitation over the Tibetan Plateau during 1979 and 2015, which was produced by using the atmospheric reanalysis product to drive the forward Eulerian moisture tracking model (WAM-2). We tracked the evaporation from global oceans ( 'oceans'  in Masks.nc), the western oceans dominated by the westerlies ( 'western oceans'  in Masks.nc), and the Indian Ocean dominated by the Indian monsoon ( 'indian ocean'  in Masks.nc),

🔍 Zoom in

and quantified their contributions to precipitation over the plateau from two aspects of absolute contribution (mm, equivalent water height in each grid) and relative contribution (%, ratio of oceanic moisture contribution to each grid). Due to date uncertainties, we used three high-resolution global atmospheric reanalysis products (ERA-I, MERRA2 and JRA55) in the moisture tracking, and we suggest the users of this dataset pay attention to the uncertainties among the different simulations.

**⚙ File naming and required software**
[Figure]

The dataset is stored in nc format. 'Oceanic moisture contribution_ERA-I.nc' is the dataset of oceanic moisture contribution simulated by WAM-2 driven by ERA-I. Therein, the variables  'oceans_*' ,  'western_oceans_*'  and  'indian_ocean_*'  represent the moisture contributions from global oceans, the western oceans, and the Indian Ocean, respectively. The variables  '*_absolute'  and  '*_relative'  represent the absolute and relative moisture contribution. For example, the variable  'indian_ocean_absolute'  represents the absolute contribution of moisture from the Indian Ocean.  'Masks.nc'  contains the mask data of the global oceans, the western oceans and the Indian Ocean used in the moisture tracking simulations.

**📖 Data Citations**          🏳 What's data citation?  📋 Data citation guideline

**Cite as:**

Li, Y. (2022). **Dataset of oceanic moisture contribution to precipitation over the Tibetan Plateau simulated by WAM-2 during 1979-2015**. National Tibetan Plateau Data Center, DOI: 10.11888/Atmos.tpdc.272946. CSTR: 18406.11.Atmos.tpdc.272946. (Download the reference: RIS | Bibtex )

**Related Literatures:**

1. Li, Y., Wang, C.H., Huang, R., Yan, D.H., Peng, H., & Xiao, S.B. (2023). Spatial Distribution of Oceanic Moisture Contribution to the Precipitation over the Tibetan Plateau. Hydrology and Earth System Sciences.( View Details | Bibtex)

🚫 Using this data, the data citation is required to be referenced and the related literatures are suggested to be cited.

**🗐 Support Program**

Second Tibetan Plateau Scientific Expedition Program

**⚒ Copyright & License**

To respect the intellectual property rights, protect the rights of data authors,expand servglacials of the data center, and evaluate the application potential of data, data users should clearly indicate the source of the data and the author of the data in the research results generated by using the data (including published papers, articles, data products, and unpublished research reports, data products and other results). For re-posting (second or multiple releases) data, the author must also indicate the source of the original data.

Example of acknowledgement statement is included below: The data set is provided by National Tibetan Plateau Data Center (http://data.tpdc.ac.cn).

License :
[Figure]

[Figure]
 This work is licensed under an Attribution 4.0 International (CC BY 4.0)

**⦉ Related Resources**

Related datasets   Service records   Recommendations

1.Distributions of debris flows in CPEC and Tianshan Mountain

2.Modeling results of leaf area index in the middle and lower Heihe River Basin at the north of Qilian Mountains (2001-2015)

3.Driving data of surface meteorological elements in the eastern Qinghai Tibet Plateau with a horizontal resolution of 3km * 3km and an hour (2010)

4.Namuco station (2019) and Southeast Tibet station (2021) air pollutant flux and vertical gradient data set

5.10 m meteorological gradient data set of hulugou basin (2012)

6.Landslide data set of Three Rivers Basin in the southeast of Qinghai Tibet Plateau

7.30 m land cover classification product data set of Qilian Mountain Area in 2021 (V3.0)

8.HiWATER: Dataset of flux observation matrix (eddy covariance system of Zhangye wetland Station) of the MUlti-Scale Observation EXperiment on Evapotranspiration over heterogeneous land surfaces 2012 (MUSOEXE-12)

9.Precipitation observation data of the east bank of Selincuo Lake (2016-2017)

10.1:4 million geomorphic type data of Qinghai Tibet Plateau (1996)

**Comments**

Please input your comment

 verification code           Submit

Current page automatically show **English** comments  Show comments in all languages

**Sidebar**

⬇ Download

☆ Follow

**🏷 Keywords**

**Discipline:** Atmosphere
Terrestrial Surface

**Theme:** Precipitation
Moisture Source
Moisture Transport
Atmospheric Water Circulation
Atmospheric Water Vapor

**Places:** Tibetan Plateau

**Time:** 1979~2015

**⊕ Geographic coverage**

[Figure]

| East: 107.00 | West: 66.00 |
| South: 23.00 | North: 44.00 |

**ⓘ Details**

**📄 File List**

Temporal resolution: Monthly
Spatial resolution: 0.5° - 1°
File size: 28 MB
Views: 31
Downloads: 0
Access: Open Access
Temporal coverage: 1979-01-01 To 2015-12-31
Temporal coverage: January 1979-December 2015
Updated time: 2022-11-26

**📇 Contacts**

⊛ : ✉ LI Ying
Distributor: National Tibetan Plateau Data Center
Email: ✉ data@itpcas.ac.cn

**⬇ Export metadata**

**Response to Referee #2 (Prof. Ruud van der Ent)**

1. "Eulerian grids enable the WAM-2layers to consider moisture budget from precipitation and evaporation separately."
**Comments:** I think the adapted sentence is better, but it is not the Eulerian grid that enables to evaluate P and E separately. The model by Tuinenburg and Staal cited here is Lagrangian and also uses P and E. Whether a tracking model is Lagrangian or Eulerian is a feature that is unrelated to if and how P and E are treated.

**Response:** Thanks for correcting this. We have revised this sentence to "the selection of WAM-2layers enables us to consider moisture budget from precipitation and evaporation separately on Eulerian grids".

2. Table AC2 (i.e., Table S2).
**Comments:** To make it easier for the reader, please add the corresponding pressure (under standard surface pressure).

**Response:** Thanks. The corresponding pressures were added to the Table (Table S2 in in our revised Supplementary).

**Table S2.** Summary of the selected 17 model layers in three reanalysis products. The column "Pressure" represents the corresponding pressures under standard surface pressure.

|     | ERA-Interim | | MERRA-2 | | JRA-55 | |
|-----|-------------|----------------|-------------|----------------|-------------|----------------|
|     | Model layer | Pressure (hPa) | Model layer | Pressure (hPa) | Model layer | Pressure (hPa) |
| 1   | 60 | 1012.05 | 72 | 1013.25 | 1  | 998.50 |
| 2   | 59 | 1009.06 | 71 | 998.05  | 2  | 995.50 |
| 3   | 58 | 1004.64 | 70 | 982.77  | 3  | 991.50 |
| 4   | 57 | 998.39  | 69 | 967.48  | 4  | 985.50 |
| 5   | 56 | 989.95  | 68 | 952.20  | 5  | 977.00 |
| 6   | 55 | 979.06  | 67 | 936.91  | 6  | 966.00 |
| 7   | 54 | 965.57  | 66 | 921.63  | 7  | 953.00 |
| 8   | 51 | 908.65  | 65 | 906.34  | 9  | 917.98 |
| 9   | 48 | 828.05  | 61 | 845.21  | 12 | 846.96 |
| 10  | 47 | 796.59  | 59 | 809.56  | 14 | 786.96 |
| 11  | 44 | 691.75  | 55 | 707.70  | 17 | 684.41 |
| 12  | 41 | 573.38  | 52 | 605.88  | 20 | 571.90 |
| 13  | 38 | 461.90  | 49 | 491.40  | 23 | 458.38 |
| 14  | 35 | 353.23  | 46 | 377.07  | 26 | 351.86 |
| 15  | 32 | 257.36  | 44 | 288.93  | 29 | 257.36 |
| 16  | 27 | 132.76  | 40 | 150.39  | 34 | 132.88 |
| 17  | 17 | 18.81   | 28 | 19.79   | 44 | 18.99  |

3. "Considering the size of the data, we will make the data that support the findings of this study available upon reasonable request."
**Comments:** I still see no reason not to make the data generated available in an open data

repository. The data holding the content of the figures/tables should not be more than a few megabytes. Moreover, there are many data repositories that can hold terrabytes of data free of charge.

**Response:** Thanks for the suggestion. We have uploaded the generated data to the National Tibetan Plateau Data Center (TPDC) (available at https://data.tpdc.ac.cn/en/disallow/c6f758cf-6c99-4023-8026-f59e6d3657cb/). Please also see our response to Executive editor above.

4. "In addition, we have detailed the datasets and the code of WAM-2layers used in this work in the Data Availability section."
**Comments:** I may be mistaken and perhaps the editor can clarify this issue further, but my point was that the data availability should indicate where the data you generated is availabe and NOT where the data that you used is availabe. The latter can be described in methods/acknowledgement and references.

**Response:** Thanks. We have revised this section to add the link/reference for the dataset generated from this work, which is now archived at the National Tibetan Plateau Data Center (TPDC): "The results of oceanic moisture tracking simulations are archived at the National Tibetan Plateau Data Center (TPDC): http://data.tpdc.ac.cn/en/data/c6f758cf-6c99-4023-8026-f59e6d3657cb/ (Li, 2022)."
Reference: Li, Y.: Dataset of oceanic moisture contribution to precipitation over the Tibetan Plateau simulated by WAM-2 during 1979-2015, Archived at the National Tibetan Plateau Data Center, available at: https://doi.org/10.11888/Atmos.tpdc.272946, last access: 29 November 2022.

**Response to Polina Shvedko:**

Please ensure that the colour schemes used in your maps and charts allow readers with colour vision deficiencies to correctly interpret your findings. Please check your figures using the Coblis – Color Blindness Simulator (https://www.color-blindness.com/coblis-color-blindness-simulator/) and revise the colour schemes accordingly.

**Response:** Thanks for the reminder. We have checked all our figures using Coblis and found that Figures 5, 6, 7, and S1 are not color-blind-friendly. We have changed the color schemes of these four figures.